# Reduced and unstratified crust in CV chondrite parent body

Clément Ganino [1] & Guy Libourel[2]

Early Solar System planetesimal thermal models predict the heating of the chondritic protolith and the preservation of a chondritic crust on differentiated parent bodies. Petrological and geochemical analyses of chondrites have suggested that secondary alteration phases formed at low temperatures (<300 °C) by fluid-rock interaction where reduced and oxidized Vigarano type Carbonaceous (CV) chondrites witness different physicochemical conditions. From a thermodynamical survey of Ca-Fe-rich secondary phases in CV3 chondrites including silica activity ($a SiO_2$), here we show that the classical distinction between reduced and oxidized chondrites is no longer valid and that their Ca-Fe-rich secondary phases formed in similar reduced conditions near the iron-magnetite redox buffer at low $a SiO_2$ ($\log(a SiO_2)$ $<-1$) and moderate temperature (210–610 °C). The various lithologies in CV3 chondrites are inferred to be fragments of an asteroid percolated heterogeneously via porous flow of hydrothermal fluid. Putative 'onion shell' structures are not anymore a requirement for the CV parent body crust.

[1] Université Côte d'Azur, CNRS, OCA, IRD, Géoazur, 250 rue Albert Einstein, Valbonne 06560, France. [2] Université Côte d'Azur, CNRS, OCA, Lagrange, 96 Boulevard de l'Observatoire, Nice 06000, France. Correspondence and requests for materials should be addressed to C.G. (email: ganino@unice.fr)

Few chondritic meteorites still preserve a pristine record of the physicochemical processes that occurred in the early solar accretion disk via their primordial components, i.e., presolar grains, calcium aluminum-rich inclusions, chondrules, or matrices[1]. Instead, the great majority has been affected by secondary fluid-assisted thermal processes occurring after accretion in their asteroidal parent bodies (refs [2, 3] for a review). Despite their obliterating role, these secondary processes shed light on chondrite parent body internal structure and dynamics and their timing provides important constraints on the accretion ages of these bodies, the first building blocks of our planetary system.

In addition to mineralogical observations, thermodynamic analysis, and oxygen-isotope compositions[3] recent $^{53}Mn$–$^{53}Cr$ ages of secondary minerals (e.g., fayalite) confirmed an early accretion age of the CV and CO parent bodies of about 1.8–2.5 Myr after CV CAIs[4] and provides compelling evidence that these secondary minerals formed in an asteroidal setting[3–5]. CV3 chondrites possessing a unidirectional magnetization have been thus interpreted as the unmelted crusts of internally differentiated early planetesimals heated primarily by the short-lived radioisotope $^{26}Al$[6, 7]. The preservation of a conductively cooled chondritic crust on the CV-differentiated parent body likely requires a progressive heating (and devolatilization?) of the CV chondritic protolith that should be recorded in the thermal and redox conditions of formation of the secondary phases present in the CV chondrites of our collections.

Here, our objective is to ascertain the physicochemical conditions of the formation of the secondary phases in the different CV chondrites, in order to check the consistency of the proposed evolution of the carbonaceous chondrites as the crust of a partially differentiated body[7–9]. The ubiquitous occurrence of Ca-Fe-rich secondary phases, e.g., andradite (adr), hedenbergite (hd), kirschsteinite (kst), wollastonite (wo), fayalite (fa), Fe ± Ni metal, sulfides, and/or Fe-oxides, in the CV chondrite subgroups ($CV_{Red}$, $CV_{OxA}$, $CV_{OxB}$) provides motivation for this study. Indeed, there are still uncertainties associated with the temperature, the oxygen fugacity, and the nature of the fluid and its composition prevailing during secondary thermal processes in different carbonaceous chondrites[2, 3, 10]. Further, the intricacies of the physicochemical processes between aqueous alteration and Fe-alkali-halogen metasomatism still remain debatable[3].

## Results

### Survey of secondary Ca-Fe-rich phases in CV3 matrices.
Following the classifications of McSween[11], CV3 chondrites were historically divided into oxidized ($CV_{Ox}$) and reduced ($CV_{Red}$) subgroups based on their bulk chemistry and their secondary mineralogy, especially the presence or absence of magnetite. Although, recent X-ray[12] and Mossbauer[13] characterizations have shown that magnetite abundance alone is not always consistent with this classification. Weisberg et al.[14] further subdivided the oxidized subgroup into the Bali-like ($CV_{OxB}$) and Allende-like ($CV_{OxA}$) types in part because of different chondrule/matrix ratios and oxygen isotope composition but mostly on the basis of their secondary mineralogy again. The Bali-like meteorites are characterized by the presence of nearly pure fayalite (Fa > 95), which is rare in the $CV_{Red}$ and largely lacking in the $CV_{OxA}$ chondrites, and the relatively high abundance of hydrous minerals, even if hydrous minerals and Fe-oxides on which these classifications are based never exceed 5–6 vol% in the CV3[12]. Our survey of mineralogy in the CV3 subgroups confirms the ubiquitous occurrence of Ca-Fe-rich assemblages (see Fig. 1 and Table 1). Ca-Fe-rich pyroxenes (i.e., hedenbergite) are by far the most dominant secondary phases and are frequently found in

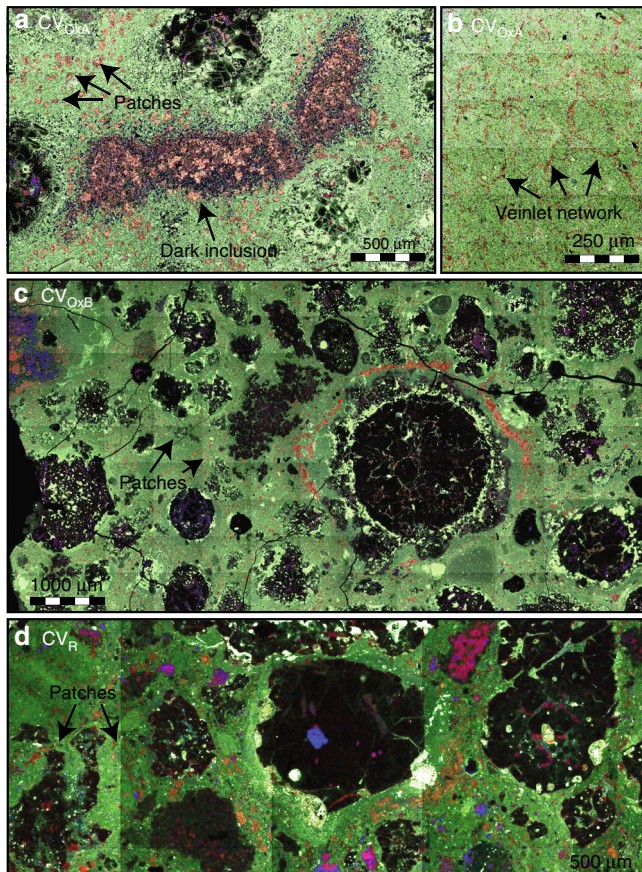

**Fig. 1** Occurrence of secondary phases in representative CV3 lithologies. Backscattered electron images coupled with energy-dispersive X-ray chemical maps (*red* = calcium; *blue* = aluminum; *green* = iron) of **a**, **b** Allende $CV_{OxA}$, **c** Bali $CV_{OxB}$, **d** Vigarano $CV_{Red}$. Notice the pervasive ubiquitous occurrence of secondary Ca-Fe-rich phases (in *pink*) in the different lithologies: **a** an andradite-hedenbergite-rich matrix and dark inclusion in Allende, **b** a veinlet network of hedenbergite in a dark inclusion in Allende, **c** andradite-hedenbergite-rich matrix in Bali, **d** kirschsteinite-hedenbergite-rich matrix in Vigarano

association with andradite (only in $CV_{Ox}$), wollastonite, and/or kirschsteinite[15, 16] (abundant in $CV_{Red}$, rare but present in $CV_{Ox}$). Calcium-rich phases: larnite ($Ca_2SiO_4$) and rankinite ($Ca_3Si_2O_7$), indicative of silica-depleted conditions, were also documented as inclusions in andradite in Bali. Finally, we confirm the occurrence of both FeNi metal and sulfides in the reduced and oxidized subgroups[15].

Our observations on the representative CV3 samples, i.e., Vigarano ($CV_{Red}$), Bali ($CV_{OxB}$), and Allende ($CV_{OxA}$), show that matrix consists largely of abundant iron-rich olivine ($Fa_{45–55}$), 1–10 μm in length, in some cases hosting minute inclusions of iron sulfides, hercynitic spinel and voids, and frequently associated with pyrrhotite/pentlandite sulfides. The secondary minerals include abundant Ca-Fe-rich silicates, Fe-oxides, sulfides, Fe-Ni-Co metal, Na-Al-Cl-rich feldspathoids, and phyllosilicates. Secondary Ca-, Fe-rich silicates are abundant not only in the $CV_{Ox}$ chondrites but also in the $CV_{Red}$ chondrites (Table 1), suggesting that the major physical and chemical process responsible of their formation occurred widely in the CV parent bodies.

Hedenbergite-dominated Ca-Fe secondary phases in CV3 subgroups occur under three similar types of textural setting (Fig. 1): micron-sized porous and polycrystalline

**Table 1 Survey of secondary minerals in CV3 chondrites matrices**

| | $CV_{OxA}$ Allende | $CV_{OxB}$ Bali | $CV_{OxB}$ Kaba-mokoia | $CV_{Red}$ Vigarano | $CV_{Red}$ Efremovka |
|---|---|---|---|---|---|
| *Phases* | | | | | |
| Fayalite $Fe_2SiO_4$ | | ++*** (g) | ++ (a,e,f) | | |
| Kirschsteinite $CaFeSiO_4$ | _(a) | | _(e) | ++(b) | +(b,e) |
| Andradite $Ca_3Fe_2Si_3O_{12}$ | ++(a) | ++(this study) | ++(a,e) | | _**(a,e,l) |
| Hedenbergite $CaFeSi_2O_6$ | ++(a) | ++(this study) | ++(a,e) | +(b) | +(b,e) |
| Wollastonite $CaSiO_3$ | _(a,c) | +(c,this study) | | | _(e) |
| Larnite $Ca_2SiO_4$ | | _(this study) | | | |
| Rankinite $Ca_3Si_2O_7$ | | _(this study) | | | |
| Magnetite $Fe_3O_4$ | _(a) | +(this study) | +(a,e) | _*(i) | |
| Troilite FeS | | | | +(c) | +(c) |
| Pentlandite $(Fe,Ni)_9S_8$ | +(c) | +(c) | +(c) | | |
| Awaruite $FeNi_3$ | +(c) | +(c) | +(c) | +Low-Ni metal (c) | |
| Cohenite $(Fe,Ni)_3C$ | | _***(k) | | | |
| Anthophyllite-Talc $Mg_3Si_4O_{10}(OH)_2$ | _(d) | | | | |
| Serpentine $(Mg,Fe)_3Si_2O_5(OH)_4$ | _(h) | ++(h) | ++(h) | ++(h) | _(h) |
| Nepheline $Na_3KAl_4Si_4O_{16}$ | +(a) | +(a) | +(a) | _(a) | _(a) |
| Sodalite $Na_8(Al_6Si_6O_{24})Cl_2$ | +(a) | +(a) | +(a) | _(a) | _(a) |
| *Assemblages* | ++hd/adr (a,b,c) | ++hd/fa (g) | ++fa/hd (a) | ++hd/kst/ ± Fe-Ni-metal (b) | ++hd/kst (b) |
| | +adr/wo (c) | ++hd/adr (this study) | ++hd/adr (a) | +Fe-Ni-metal/mt* (i) | _hd/wo (e) |
| | +kst/wo (a) | +hd/wo (this study) | +hd/wo (e) | | _hd/adr (e,l) |
| | _hd/kst (a,b,c) | +aw/pn/ ± mt (c) | +aw/pn/ ± mt (j) | | |
| | +aw/pn/ ± mt (j) | _adr/rnk or lrn (this study) | _hd/kst (e) | | |
| | | _fa/mt/chn/Fe-Ni-metal (k) | | | |

(a) Brearley and Krot (2012); (b) MacPherson and Krot (2015); (c) Krot et al. (1998); (d) Brearley (1997); (e) Krot et al. (1997); (f) Hua and Buseck (1995); (g) Jogo et al. (2009); (h) Krot et al. (2006); (i) Abreu and Brearley (2011); (j) Blum et al. (1989); (k) Krot and Todd (1997); (l) Biryukov et al. (1998)
++frequent; +common; —rare; * in chondrules; **in a clast; *** in 'Bali-like' clasts in Vigarano

nodules or patches scattered in the iron-rich olivine dominated matrix, fine polycrystalline veinlets, forming in some cases an imbricated network in the matrix or in dark inclusions, and larger delineated area (veins or dark inclusions) showing radial or lateral mineralogical zoning. In the former case, secondary phases formed mineralogical zoned concentric patches of several tens of microns in diameter showing imbricated associations of Ca-Fe pyroxene and/or andradite, with or without nepheline and/or sodalite. In the other settings, a mineral zoning still exists but parallels the boundaries of the vein or the dark inclusion with the matrix. Irregular shaped hedenbergite ± andradite or hedenbergite ± kirschsteinite assemblages form veins in matrix that surround not only CAI or dark inclusions[15, 16] but also chondrules (Fig. 1). Frequently these assemblages insulate a discontinuous porous central part made of voids or elongated cavities[17] filled with euhedral Ca-Fe crystals. Finally, it worth noticing that the CV chondrites contain FeNi metal and sulfides in the reduced and oxidized subgroups (this study and ref. [18]).

There is an historical agreement in the cosmochemist community on the fact that the reduced and oxidized assemblages in CV3 chondrites formed under different physicochemical conditions[5, 16]. On the basis of the thermodynamical modeling of phase equilibrium among mineral phases and the oxygen isotopic composition, it has been further suggested that secondary phases resulted from low temperatures (<300 °C) fluid-rock interaction in an asteroidal setting[4, 15, 19, 20].

Main occurrences of secondary Ca-Fe-rich minerals are summarized in Table 1. The reader is referred to more specific papers for detailed information on the different CV[3, 5, 15, 21–23].

In $CV_{OxA}$, documented here with Allende (Supplementary Fig. 1), the matrix consists largely of abundant lath-shaped, elongate, iron-rich olivine ($Fa_{45–55}$), 1–10 μm in length, hosting minute inclusions of iron sulfides, hercynitic spinel and voids, and frequently associated with pyrrhotite/pentlandite sulfides.

The secondary minerals include abundant Ca-Fe-rich pyroxenes belonging to the diopside-salite-hedenbergite solid solution, andradite, wollastonite, and abundant nepheline and sodalite. We did not observed pure fayalite and kirschsteinite in this study, the later being documented in the litterature[3]. In agreement with previous studies, we only found scarce phyllosilicates, questioning the interpretation that Allende may have been affected quite extensively by aqueous fluids on an asteroidal parent body.

It is generally admitted that the $CV_{OxB}$ chondrites (e.g., Kaba, Bali) experienced aqueous alteration resulting in the replacement of primary minerals in chondrules, CAIs and ameboid olivine aggregates by secondary phyllosilicates, magnetite, Fe,Ni-sulfides, Fe,Ni-carbides, fayalite, salite-hedenbergite pyroxenes, and andradite. This is what we observed in our sample (Supplementary Fig. 1). As calcium bearing silicates, we also documented larnite ($Ca_2SiO_4$) and rankinite ($Ca_3Si_2O_7$), both occurring as inclusion in andradite. Our observations do not document fayalite[24] neither abundant phyllosilicates[25] as generally described. Sodalite is a very common other secondary, sometimes closely associated with andradite or hedenbergite.

In our last case of $CV_{Red}$ chondrite, our reference sample of Vigarano, (Supplementary Fig. 1) contains clasts of the $CV_{OxB}$ and $CV_{OxA}$ materials[26], which were not the objectives of our work. Pure fayalite (Fa99) is found as clast and associated to magnetite in the matrix. The matrix also contains abundant secondary Ca-Fe phases (frequent kirschsteinite and hedenbergite) that display the same type of occurrence than secondary Ca-Fe silicates in $CV_{OxA}$ and $CV_{OxB}$. In both Leoville and Vigarano, kirschsteinite solid solution ($Kir_{82–94}Mon_{1–15}$) and Ca-, Fe-rich pyroxene coexist together in clumps, with Fe-Ni metal enclosed within the kirschsteinite[27]. Frequently, hedenbergite and/or kirschsteinite are zoned from Fe-rich endmember in the crystal core to a more Mg-rich composition in the crystal rim[16, 27, 28].

Metal in the $CV_{Red}$ and $CV_{OxB}$ chondrites is mainly low-Ni, while high-Ni awaruite ($FeNi_3$) dominates in the $CV_{OxA}$ chondrites. In the $CV_{OxB}$ lithologies, the secondary minerals include magnetite, Ni-rich metal (awaruite), and sulfides, whereas in the $CV_{OxA}$ lithologies the secondary minerals include magnetite, Ni-rich sulfides, Ni-and Co-rich metal (awaruite and wairauite). In$CV_{Red}$, the matrix contain Fe,Ni-metal and sparse Fe,Ni sulfide, but no magnetite except at the periphery of type IA chondrules in Vigarano[29]. Hydrous phyllosilicates are almost absent in both the $CV_{Red}$, rare in $CV_{OxA}$ chondrites (up 2 vol%) and slightly more common in the $CV_{OxB}$ chondrites in which their modal abundances rise up to 5 vol%[12]. $CV_{Red}$ and $CV_{OxB}$ chondrites are dominated by saponite with only rare serpentine[30]. Neither saponite nor serpentine has been reported from Allende, only talc and amphibole (tremolite with minor anthophyllite) have been described as hydrated alteration products[31].

**Silica activity: an overlooked key parameter**. That metal, oxides, and silicates hosting both metallic, ferrous, and ferric iron may coexist clearly suggests that oxygen fugacity plays a significant role in determining the composition and the occurrence of Ca-Fe-rich secondary phases in CV chondrite matrices. This view is supported by paragenetic relations obtained experimentally by Gustafson[32] in the system Ca-Fe-Si-O and his Schreinemaker's analysis, where the stability field of andradite must be confined to relatively oxidizing conditions well above the Fayalite-Magnetite-Quartz (FMQ), while hedenbergite (and/or kirschsteinite) may extend their stability field down to more reducing conditions. Both single phases: andradite and/or hedenbergite, are thus stable over a range of log $fO_2$ and $T$, and so their conditions of formation are poorly constrained (Fig. 2). Nevertheless, this analysis provides a consistent thermodynamic framework for the present subdivision of CV3 chondrites into andradite-bearing oxidized ($CV_{Ox}$) and kirschsteinite-bearing reduced ($CV_{Red}$) subgroups[16] and their inferred different conditions of formation[5].

We challenge this classical view by noticing that the low silica activity of CV3 chondrites, as suggested by the noteworthy absence of silica phases in their matrices, is an overlooked key parameter controlling the stability field of the Ca-Fe-rich secondary phase assemblages. Indeed, if we examine andradite in a Ca-Fe-Si-O system, its stability relative to hedenbergite[32, 33] is governed by the reaction:

$$3\,Ca_3Fe_2Si_3O_{12(adr)} + 9\,SiO_{2(silica)} + Fe_3O_{4(mt)} = \atop 9\,CaFeSi_2O_{6(hd)} + 2\,O_{2(fluid)} \tag{1}$$

In a silica-saturated system involving pure phases, this equilibrium lies between the Hematite-Magnetite and FMQ buffers at about two log units above FMQ as aforementioned (Fig. 2). We note, however, that in a silica-undersaturated system with pure phases, the equilibrium constant for reaction (1) is

$$\log K_{(1)} = 2\log fO_2 - 9\log aSiO_2 \tag{2}$$

Solving for log $fO_2$ gives then

$$\log fO_2 = 1/2\log K_{(1)} - 9/2\log aSiO_2 \tag{3}$$

and the shift in the equilibrium log$fO_2$ due to degree of silica undersaturation is

$$\Delta_{K_{(1)}}\log fO_2 = +9/2\log aSiO_2 \tag{4}$$

Changes in silica undersaturation causing the term log $aSiO_2$ to be negative shift then significantly ($-4.5$ log units $fO_2$ shift per order of magnitude of $aSiO_2$) the equilibrium location of reaction (1). Thus, the effect of the silica activity is significant and

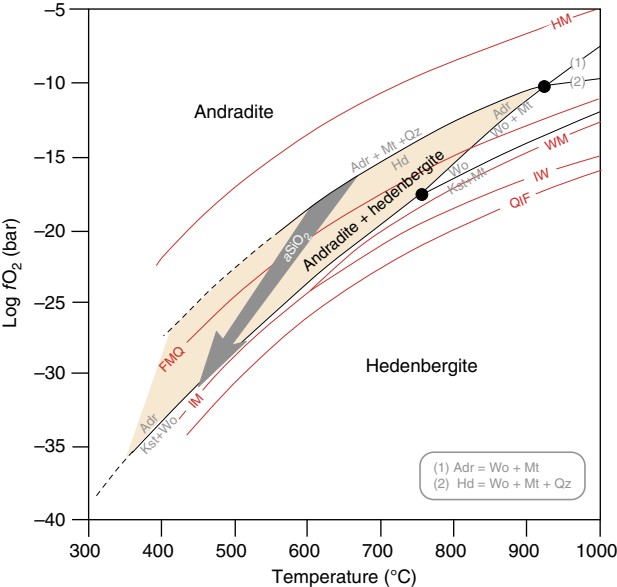

**Fig. 2** Oxygen fugacity vs. temperature plot of the stability field of hedenbergite and andradite. Both single phases: andradite and/or hedenbergite, are stable over a range of log $fO_2$ and $T$, and poorly constrained their conditions of formation[17]. Nevertheless, this thermodynamic framework is consistent with the present subdivision of CV3 chondrites into andradite-bearing oxidized ($CV_{Ox}$) and kirschsteinite-bearing reduced ($CV_{Red}$) subgroups and their inferred different conditions of formation. When the two phase assemblage andradite + hedenbergite is considered, the redox conditions of formation of the Ca-Fe-rich secondary phases could be a bit more restricted but the temperature remains still poorly constrained. Low silica activity of CV3 chondrites is an overlooked key parameter controlling the stability field of the Ca-Fe-rich secondary phase assemblages: the *gray arrow* indicates the effect of decreasing the silica activity of the system on the stability field of andradite

should have profound implications on the phase relationships in the Ca-Fe-Si-O system.

We calculated the stability fields in the Ca-Fe-Si-O system as functions of log $aSiO_2$ – log $fO_2$ – $T$ at an indicative pressure of 2000 bars, between 100 and 900 °C, and plausible ranges of $fO_2$ and $aSiO_2$. The bulk composition of Allende from Jarosewich[34] simplified to its Ca-Fe-Si-O components was used as a representative input composition. Visual inspection of calculated log $aSiO_2$ – log $fO_2$ isothermal sections at 300 and 500 °C (Fig. 3) shows how silica-saturated phases (e.g., quartz and ferrosilite) and silica-undersaturated phases, (e.g., hedenbergite, kirschsteinite, rankinite, and larnite) are located according to their silica activity potential. Clearly, as the silica activity of the system decreases, the andradite stability field (reaction (1)) shifts toward lower oxygen fugacity and, at moderate temperatures ($\approx$350 °C), encounters the iron-magnetite reaction curve (IM), which is independent of $aSiO_2$. Such a trend contrasts, for instance, with the appearance of hedenbergite in reduced conditions in the reaction

$$3\,CaSiO_{3(wo)} + SiO_{2(silica)} + Fe_{(metal)} + \atop 1/2\,O_{2(fluid)} = CaFeSi_2O_{6(hd)} \tag{5}$$

for which the opposite effect is expected—the hedenbergite stability field shifts toward higher oxygen fugacities as the silica activity of the system is decreased

$$\Delta_{K_{(5)}}\log fO_2 = -2\log aSiO_2 \tag{6}$$

These log $aSiO_2$ – log$fO_2$ relationships can be generalized for

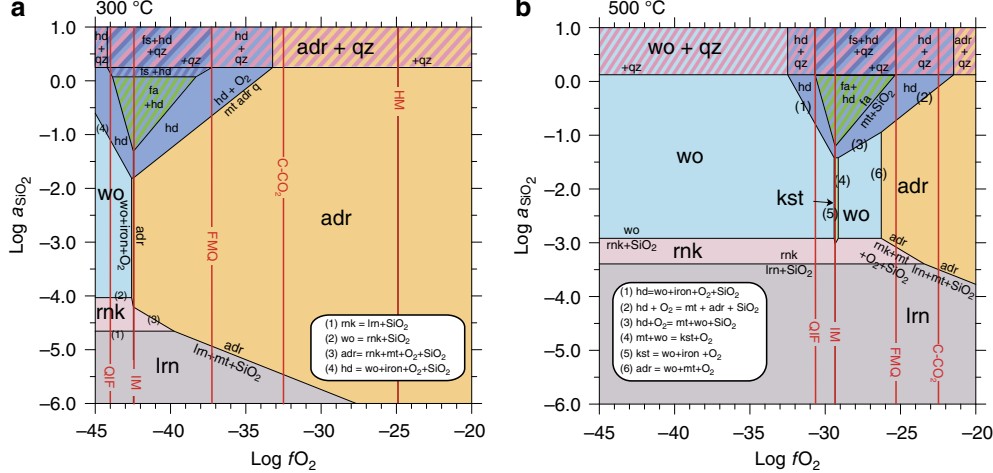

**Fig. 3** Silica activity vs. oxygen fugacity plot of stable phases in the system Ca-Fe-Si-O at **a** 300 °C and **b** 500 °C at an indicative pressure of 2000 bar. Kirschsteinite is stable for $T > 450$ °C, relatively low silica activity and in IM-buffered conditions. Andradite is stable both in oxidative and reductive condition depending the temperature (stable in IM-buffered condition for $T < 350$ °C)

different reactions involving Ca-Fe secondary phases in silica-undersaturated systems as follows:

$$CaSiO_{3(wo)} + SiO_{2(silica)} + 1/3\,Fe_3O_{4(mt)} = \\ CaFeSi_2O_{6(hd)} + 1/3\,O_{2(fluid)} \qquad (7)$$

$$\Delta_{K_{(7)}} \log fO_2 = +3 \log aSiO_2 \qquad (8)$$

$$3\,Fe_2SiO_{4(fa)} + 1/2\,O_{2(fluid)} = SiO_{2(silica)} + 1/3\,Fe_3O_{4(mt)} \qquad (9)$$

$$\Delta_{K_{(9)}} \log fO_2 = +3 \log aSiO_2 \qquad (10)$$

$$Fe_2SiO_{4(fa)} = SiO_{2(silica)} + 2\,Fe_{(metal)} + O_{2(fluid)} \qquad (11)$$

$$\Delta_{K_{(11)}} \log fO_2 = -1 \log aSiO_2 \qquad (12)$$

This new finding has major implications: occurrences of hedenbergite, kirschsteinite, rankinite, and larnite in that order in different assemblages require significant decreases in the silica activity of the system. Kirschsteinite is stable only in reduced (IM buffered), low silica activity (log $aSiO_2 \ll -1$), and high-temperature conditions (T > 450 °C). Fayalite + hedenbergite ($\pm$ magnetite) assemblages are restricted to Fe-rich and high silica activity. Stable association of fayalite with kirschsteinite ($\pm$ hedenbergite) is possible only at high-temperature. If andradite is stable at high temperature in oxidizing conditions (Fig. 3b), it noticeably expands its stability field towards reducing conditions at low temperature in low silica activity environments ($\approx$ log $aSiO_2 < -1.7$); andradite being co-stable with iron and magnetite below 350 °C. That the stability of andradite is highly sensitive to silica activity explains therefore why, in rocks of low silica activity, andradite can be stable even in the presence of native iron (Fig. 3a, see also ref. [35]).

**Reduced redox conditions for CV3 parent body metamorphism.** Ca-Fe secondary phase stability fields in the CV3 chondrites can be then interpreted in the light of this new thermodynamic frame as follows. The occurrence in $CV_{OxA}$ of clusters of awaruite $\pm$ pentlandite $\pm$ magnetite[3, 36–38], the occurrence in $CV_{OxB}$ of Fe-Ni sulfide $\pm$ magnetite $\pm$ fayalite $\pm$ hedenbergite assemblages

either as patches in the matrix or as veins destabilizing pristine Fe-rich metal blebs[3], and the occurrence in $CV_{Red}$ of hedenbergite, kirschsteinite, and Fe metal associations (e.g., ref. [27].) hint all in favor of redox conditions buffered at or very closed to the IM buffer curve for the formation of Ca-Fe-rich secondary phases in CV3 chondrites, as already suggested for the formation of the opaque assemblages in the same chondrite group[36] (see also refs [37, 39]).

Thus, a log $aSiO_2$ –T diagram (Fig. 4a) can be generated at such IM reducing conditions. It depicts clearly that andradite is a low temperature (T < 350 °C) phase, kirschsteinite is a high temperature (>450 °C) phase, wollastonite is intermediate, and fayalite and/or hedenbergite, despite being stable over a large range of temperature, are indicative of higher silica activity; in addition, fayalite is more stable in an iron-rich system. In such a frame, it is remarkable that a single trend can account for the different Ca-Fe secondary phase assemblages observed in all CV3 chondrite lithologies. Incidentally, this suggests a general evolution governed by a silica activity depletion as the temperature of the system is decreased. Changes in silica activity required for the formation of Ca-Fe secondary phases in CV3 chondrites are significant between $-2 <$ log $aSiO_2 < -1$ if only the major phases (i.e., fayalite, kirschsteinite, wollastonite, and andradite buffered by hedenbergite) are considered and between $-6 <$ log $aSiO_2 < -1$ if the minor phases (i.e., rankinite and larnite from the CV3 Bali) are included. The temperature range is more difficult to estimate but a conservative range from 210 to 610 °C is inferred from this survey of CV chondrites considering the lack of xonotlite replacing wollastonite and the occurrence of pentlandite, respectively. The absence of awaruite ($T_{max} = 520$ °C[40]) and pentlandite ($T_{max} = 610$ °C[41]) and the occurrence of kirschsteinite ($T_{min} = 450$ °C, Fig. 4a) suggest that the kirschsteinite $\pm$ hedenbergite bearing assemblages, as observed for instance Allende, Vigarano, Leoville or Efremovka, are amongst those requiring the highest temperatures. Whereas, andradite $\pm$ hedenbergite bearing assemblages, as observed in Efremovka, Allende or Bali, require lower temperatures (<350 °C).

**Adequacy of these conditions with respect to other observables from CV3.** *Temperature.* The proposed large temperature range (i.e., <277 °C up to 1050 °C for the formation of secondary phases in CV3 chondrites) and the numerous discordant interpretations of the involved processes in the literature (see review in ref. [3].)

echo the difficulty of assigning metamorphic grades and absolute peak temperatures to the metamorphism of most CV3 chondrites, mainly because of their complex histories before, during and after accretion. Peak temperature assemblages are not fully constrained,

and may undergo partial or full retrogression to lower grade phase, especially if fluids are present. A temperature as high as >1050 °C is deduced from the ferrobustamite-augite two phase field in the Allende matrix[23]. The consideration of bulk rock noble

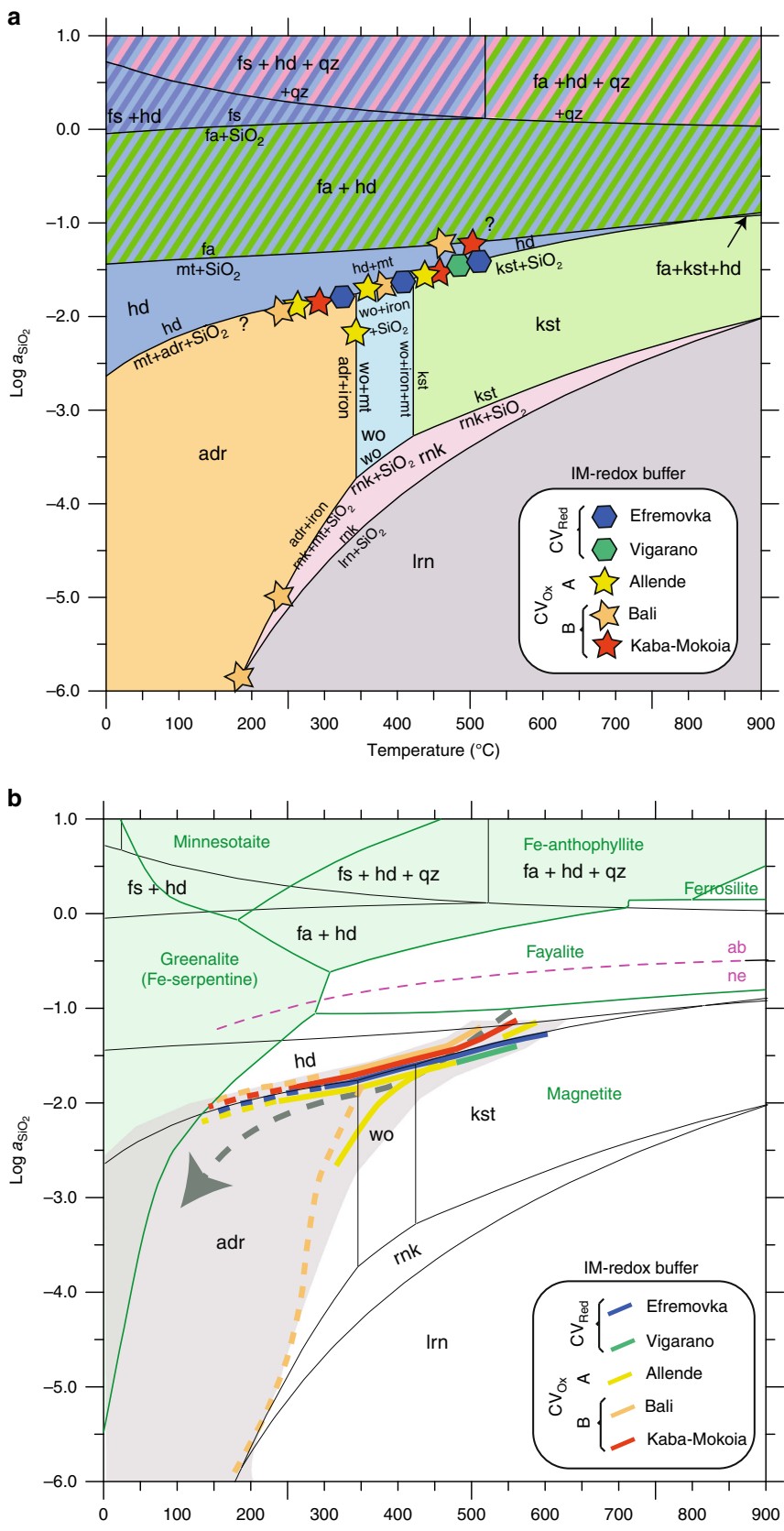

gas or presolar grain abundances also points to high-temperature process for Allende with relatively the other CV chondrites[42]. Independently to our study on inorganic phases, it is of note that Insoluble Organic Matter's sulfur speciation and structural order[43], support the formation of secondary phases in Allende parent body at temperature as high as 624 °C and reduced redox conditions. If the presence of amphibole and talc in Allende has been used to argue for temperatures >400 °C[31], occurrences of serpentine/saponite intergrowths (e.g., saponite) document lower temperatures (100–200 °C)[44]. Meanwhile, from thermodynamical calculations of precipitation from aqueous fluid, fayalite and Ca-Fe-rich minerals in CV chondrites were inferred to form at relatively low temperatures (<250–300 °C)[5, 19, 26]. If bulk rock considerations are taken into account, carbonaceous matter in CV matrices (see ref. [10] and references therein) is used to infer the petrologic grade and to sort the CV chondrites from the intensity of their thermal processing. It shows a wide range of estimates of peak temperature (e.g., Kaba 300–420 °C, Allende: 330–600 °C[45–48]). That Ca-Fe-rich phases and carbonaceous matter indicate a range of temperature extending over several hundreds of degrees suggests that thermal heterogeneity is a characteristic feature of matrix of CV chondrites. The conditions inferred for the formation of the hydrous secondary phases (anthophyllite and talc) in the chondrule mesostasis of Allende[31] at higher temperature than our estimates from CV matrices, are also consistent with the higher local silica activity imposed by the ubiquitous occurrence of olivine and low-Ca pyroxene in chondrules according to the reaction $Mg_2SiO_4$ (Fo) + $SiO_2$ (Qtz) = $Mg_2Si_2O_6$ (En) (ref. [19] and Fig. 4). As aforementioned, low-temperature alteration certainly below 150 °C[31] are required to form saponite and Fe-rich serpentine. In the light of these factual heterogeneities, we thus interpret our estimated temperature range for the formation of Ca-Fe-rich assemblages (from 210 to 610 °C) as recording only a portion of the putative thermal variation suffered by the CV3 parent body, without excluding the possibility of locally higher peak temperatures. As Ca-Fe-rich minerals does not inform on higher temperature process, the present model is not inconsistent with the classical order of metamorphic temperature inferred from presolar grain abundances and Raman spectroscopy studies of organic material, i.e., temperature increasing from Leoville to Vigarano to Allende.

*Redox.* The inferred reduced conditions near the IM buffer curve are compatible with intrinsic oxygen fugacity estimates for CV3 chondrite matrices as long as an Fe sulfide ± fayalite$_{40–60}$ ± magnetite ± Fe,Ni metal ± awaruite assemblage in presence of graphite is concerned (Supplementary Fig. 2). Intrinsic oxygen fugacity has to be in fact below C-CO$_2$ buffer curve, due to graphite saturation, close to awaruite saturation[35] and above the fayalite-iron-quartz (QIF) buffer curve in response of the silica undersaturation of CV3 chondrite matrices. In these conditions, it is worth noticing that magnetite is indicative of reduced conditions well below the FMQ buffer, not of oxidizing ones. The first discovery of cohenite: (Fe,Ni)$_3$C in association with metal—magnetite—fayalite in a Bali-like clast in the reduced CV3 chondrite Vigarano and its spontaneous formation at $T < 700$ K[49]

according to

$$15\,Fe_{(metal)} + 4\,CO_{(gas)} = 4Fe_3C_{(cohenite)} + Fe_3O_{4(magnetite)} \quad (13)$$

further supports such assertion of reduced conditions at IM. Finally, a secondary effect of low oxygen fugacity is the reduction of sulfur[50], leading the stabilization of low-sulfur sulfides (e.g. heazlewoodite) and sulfur-free metal alloys (e.g. awaruite and/or wairauite), as indicated by the occurrence of cluster of awaruite and Ni-sulfide in several matrices of CV3 (see Fig. 2 of O'Brien and Tarduno[38] for Allende). We notice however that these redox and thermal conditions for the formation of secondary Ca-Fe-rich minerals in the matrix disagree with the igneous temperature and more oxidizing conditions proposed for magnetite in chondrules[51].

*Hydrous phases.* The lack of xonotlite Ca$_6$Si$_6$O$_{17}$(OH)$_2$ replacing wollastonite ($T_{max}$ = 210 °C[32]), of ilvaite CaFe$^{2+}_2$Fe$^{3+}$Si$_2$O$_8$(OH) replacing hedenbergite ± magnetite ($T_{max}$ = 450 °C[32]), or of calcite CaCO$_3$ replacing wollastonite ($T_{max} \approx$ 300 °C[33, 34]) not only strengthen our $T \geq 210$ °C temperature scheme for the formation of Ca-Fe secondary phases in CV3 chondrites[17] but also hint for low or very low partial pressures of H$_2$O and CO$_2$ in the system, a feature also consistent with the inferred reducing conditions.

*Nepheline and sodalite.* Finally, Sodium-rich nepheline ((Na,K)$_2$Al$_2$Si$_2$O$_8$) and occasionally sodalite (Na$_4$Al$_3$Si$_3$O$_{12}$Cl) are present in matrices of CV3 chondrites, associated or not with Ca-Fe secondary phases (Fig. 1 and Supplementary Fig. 1; refs [3, 53]). Referred as evidences for an alkali-halogen metasomatism in CV3 chondrites of still an unknown nature, it is however difficult to evaluate if they crystallized prior to, simultaneously with, or clearly after the Ca-Fe secondary phases, showing the need of a dedicated study on this topic. Nevertheless, the crystallization of these Na-Al-Cl-rich feldspathoids are very good tracers of low silica activity environments (i.e., log $a$SiO$_2$ < −1; Fig. 4b), supporting once again the presented assertion.

**Crustal evolution in the CV3 parent body.** Silica undersaturation of CV3 chondrites together with their intrinsically unique reduced character have thus profound implications on the stability field of the Ca-Fe secondary phases, on which the subdivision of CV3 chondrites is based on. Accordingly, the present distinction between the CV$_{Red}$, CV$_{OxA}$ and CV$_{OxB}$ subtypes on the basis of redox considerations (e.g., presence or absence of magnetite and/or andradite) is no longer valid. Instead, we advocate that the formation of Ca-Fe secondary phases in the matrices of the different CV3 lithologies occurred at the same reduced redox conditions, which are close to the IM buffer during a unique silica-undersaturated allochemical metamorphism path that can be best approximated by a single log $a$SiO$_2$ – $T$ pathway (Fig. 4b). Decreases in $a$SiO$_2$ with respect to the intrinsic composition of unprocessed CV3 matrix (close to olivine/orthopyroxene silica activity buffer), and their correlative increases in $a$CaO and $a$FeO to match the observed Ca-Fe secondary assemblages indicate an open system behavior, in

---

**Fig. 4** Silica activity vs. temperature diagrams of mineral stability fields. **a** Mineral stability in the system Ca-Fe-Si-O associated with characteristic assemblages in CV3 chondrites; **b** Mineral stability in the systems Ca-Fe-Si-O without water (*black*) and Ca-Fe-Si-O-H with excess water. Stability field are calculated at IM redox conditions and an indicative pressure of 2000 bar. *Green lines* and *green area* for hydrous phases. The silica activity buffer curve nepheline/albite is also plotted. Characteristic Ca-Fe-rich assemblages in CV3 chondrites allow depicting Log $a$SiO$_2$–$T$ pathway for each CV3 lithologies. As illustrated in **b**, a single trend matches the diversity of Ca-Fe secondary phases in all CV3 subgroup, suggesting a unique allochemical metamorphism in CV3 chondrites parent body (*gray area* and *dashed arrow*). Ca-Fe-rich anhydrous phases crystallize first at the highest $a$SiO$_2$ due to the intrinsic composition of the unprocessed matrix while rare serpentine-like hydrous phases (Fe-serpentine field in *green*) should witness the low temperature part of the fluid-assisted metamorphic trend (*dashed arrow*)

which the CV3 unreacted matrices interact with Ca-Fe enriched pervasive fluids in a 210–610 °C temperature range. If their composition cannot be apprehended in detail from the presented data, it seems however unavoidable that such fluids must have been enriched in calcium to form hedenbergite, abundant and almost ubiquitous in the CV3 chondrites group. At the time of hedenbergite crystallization, these fluids must have been buffered by a reduced environment characterized by $CO/CO_2 \approx 1$, $H_2/H_2S \approx 5\text{x}10^3$, and $H_2/H_2O \approx 5$, if a temperature of 500 °C is considered[54]. Assuming a solar composition gas of $CO/CO_2 \approx 10^4$, $H_2/H_2S \approx 2.7\times10^4$, and $H_2/H_2O \approx 3.3\times10^3$[55], this rules out a nebular origin for these Ca-Fe-rich secondary phases as already proposed[4, 5]. Even if solubilities of these gaseous species are unknown, the likely addition of chlorine and alkalis to the system makes these hot fluids resembling pervasive, Darcy flow type, supercritical hydrothermal fluids. This assertion is also consistent with textural settings of secondary phases, mainly as scattered patches or, when the fluid is channelized, as subtle veinlet networks (Fig. 1). Alteration in CV chondrites does not appear to have been controlled by fluid-filled fractures, rather the entire sample is homogenously altered. Although debate continues regarding the degree and the scale to which CV chondrites were closed systems with respect to aqueous fluids, it is extremely unlikely that they were closed systems with respect to gases or supercritical fluids whatever their origin[56].

The simplest inferences for explaining such low silica activity environments are that the fluids formed very likely in a chondritic environment, naturally poorer in silica than their Earth analogs, and that CV3 matrices undergo the beginning of serpentinization. The low silica activity and the reducing conditions are the critical properties that produce the unusual geochemical environment, which is similar to a serpentinization front[57]. By analogy, such a process is capable of explaining the simultaneous occurrence of low-silica and reduced minerals, like andradite, magnetite and awaruite, in close association with serpentine-like hydrous phases (e.g., saponite), a feature very similar indeed with those observed in the CV3 chondrites.

On Earth, where ultramafic rocks are exposed to water at temperatures <400 °C, they inevitably undergo serpentinization reactions due to the efficiency of this process[58, 59]. The scarcity of serpentine in Allende, Vigarano, or more generally in CV3 chondrites[12] has thus to be questioned. Several keys could be invoked: a dehydration event following serpentization and erasing evidence of the hydration processus[5, 60], a short duration fluid flow preventing kinetically the progress of serpentinization[58], a 'dry' system with very low water partial pressure, and/or a high temperature (>360 °C[58]) that would thermodynamically prevent the serpentinization. While answering these questions is clearly beyond the scope of this work, we notice that our high temperature, $H_2O$-depleted reduced and low silica activity metamorphic conditions provide an alternative to the classical aqueous alteration models proposed for the CV3 chondrite parent body[19, 24, 61], in which Ca-Fe-rich pyroxenes, andradite, fayalite, phyllosilicates, and magnetite are all inferred to form at relatively low temperatures (<300 °C) in the presence of aqueous solutions. In our model, Ca-Fe-rich anhydrous phases crystallize first at the highest $a\text{SiO}_2$ due to the intrinsic composition of unprocessed matrix, while rare serpentine-like hydrous phases should witness the low temperature part of the fluid-assisted metamorphic trend (Fig. 4b).

From this perspective, fayalite overgrowths in $CV_{OxB}$ are thought to be the result of high-temperature sub-solidus crystallization rather than low-temperature aqueous precipitation[4, 19, 24, 61]. If this is correct, the very limited Fe–Mg interdiffusion boundaries[62] ($\leqslant 1 \,\mu m$) between fayalite and forsterite grains in the CV matrices should imply short periods of hydrothermal activity on the

parent body[4, 61] ($<10^4$–$10^5$ year between 500–600 °C), which could have wide-ranging implications for the asteroidal history of CV chondrites.

Both bulk rocks and dark inclusions in CV3 chondrites show a large range in their oxygen isotopic compositions ($-5.9 < \Delta^{17}O < -0.8‰$) indicating various degrees of interaction between solids and one or several fluid reservoirs. Such heavy-isotope enrichment is interpreted as resulting from low-temperature aqueous alteration associated with oxidation[63, 64]. Given the present observations of significant abundances of Ca-Fe-rich secondary phases vs. hydrous phases in the matrix (Fig. 1), our work supports instead a heavy-isotope enrichment resulting from interactions with hot and reduced hydrothermal fluids as recorded by the systematic heavy-isotope compositions, $\Delta^{17}O \approx -1‰$, of Ca-Fe-rich silicates (e.g., fayalite, andradite, and hedenbergite[3, 65] and references therein).

The various lithic fragments forming the CV3 chondrites including the presence of the $CV_{OxA}$ lithology in the reduced breccia Vigarano, the presence of the $CV_{OxB}$ and $CV_{Red}$ clasts in the Mokoia breccia, and the presence of dark lithic fragments in all CV3 subtypes and their diverse porosity and permeability[16] are in good agreement with CV3 lithologies being variably altered crustal pieces coming from one heterogeneously Darcy flow percolated asteroid. Each fragment records a part of a unique log $a\text{SiO}_2$ – log $f\text{O}_2$ – $T$ fluid-assisted allochemical metamorphic trend. A feature of Darcy flow, for example in aquifers on Earth, is that the percolation varies in different directions, such that different physiochemical environments may exist spatially and overlap in time. Meaning that, in different regions of the same crustal level, $a\text{SiO}_2$ and temperature could vary such that assemblages typical of $CV_{Ox}$ and $CV_{Red}$ form contemporaneously. These results, which are also consistent with the similar cosmic-ray exposure ages for the different CV3 lithologies[66], collectively indicate that fluid-assisted metamorphism on the CV3 parent asteroid should not be considered as a continuous, protracted event but rather as the cumulative effects of hydrothermal activity changes (see also ref. 60).

Reconciling the short-duration of reduced $H_2O$-depleted hydrothermal events with the effects of internal heating by the most likely heat source in primitive planetesimals, decay of $^{26}Al$ ($t_{1/2} = 0.7$ Ma), is certainly a challenge but plausible in a context of internally active pristine parent body and the short duration of supercritical fluids or vapors migration by buoyancy-driven Darcy flow[56]. Despite paleomagnetic evidence for differentiated asteroidal interiors, a stratified 'onion shell' structure as envisioned for the crust of CV chondrite parent body, e.g., with the least metamorphosed, brecciated, and reduced CV chondrites ($CV_{Red}$) nearest the surface, beneath the Bali-type oxidized CV chondrites ($CV_{OxB}$), and at greater depth, the Allende-type oxidized CV chondrites ($CV_{OxA}$)[7], is not anymore a requirement.

## Methods

**Origin of the samples**. A polished thin section of Allende, provided by the Museum National d'Histoire Naturelle (MNHN), Paris, and containing a massive Dark Inclusion about 0.35 and ~1.5 mm long (Fig. 1a) was compared with others in a massive sample of Allende provided by the Statens Naturhistoriske Museum (Copenhagen) (Fig. 1b). The sample of Bali (Fig. 1c) and Vigarano, were respectively provided by the MNHN and by the Smithsonian National Museum of Natural History (Washington, DC).

Mineral characterization: In addition to the figure presented in the main text (Fig. 1), detailed textural and mineralogical characterizations (Supplementary Fig. 1a–i) were performed with a FEG-SEM JEOL 7000 F at CRHEA CNRS-Nice (France) operating at 15–20 kV accelerating voltage and 1–2 nA beam current.

**Petrological survey**. A detailed mineralogical and petrological survey has been undertaken on four different samples from three CV3 chondrites: Allende ($CV_{OxA}$; two samples), Bali ($CV_{OxB}$), and Vigarano ($CV_{Red}$). The general overview with coupled backscattered electron images and energy-dispersive X-ray spectroscopy

chemical maps (Fig. 1 and Supplementary Fig. 1) was acquired at CEMEF Mines ParisTech-Nice (France) with a MEB FEI XL30 ESEM LaB6 operated at 20 kV and 200 nA beam current, equipped with a BRUKER Quantax 655 detector with XFlash 6|30 technology silicon drift 10 mm$^2$ at 129 eV (100 k.c.p.s.). BRUKER Micro-analysor QUANTAX was associated with the software ESPRIT (semi-quantitative analyses without standard by P/B-ZAF method).

**Thermodynamical analyses**. Thermodynamical analyses and equilibrium phase assemblage diagrams were computed using the Domino program from the Theriak-Domino software[67] and the internally consistent thermodynamic data sets from Holland and Powell[68] extended with kirschsteinite properties. Data for kirschsteinite were added to the thermodynamic database keeping its self-consistent properties as following: the entropy was calculated with the method of Holland[69], using the volume of 5.239 J bar$^{-1}$[70]. Heat capacity was calculated as proposed by Berman and Brown[71] and translated in the formalism of Holland and Powell[72]. The thermal expansion—not critical for the range of pressure investigated in this study—was supposed identical to the one of monticellite. Enthalpy was estimated from the enthalpy of monticellite corrected by a Fe-Mg substitution. This estimation (−1903.9 kJ mol$^{-1}$) was decreased by 20 kJ mol$^{-1}$ to fit the results of Markl et al.[73], who showed that the reaction hd = kst + qz occurs at 400 °C at 1 kb and log(aSiO$_2$) = −0.5.

In such modeling, the calculated stable mineral assemblage is a combined function of the selected bulk chemical composition (X) of a given volume and the prevailing conditions (P, T) during crystallization. The results presented here used Allende bulk composition simplified to its Ca-Fe-Si-O components as input. Computations using bulk composition of other CV3 lithologies (including bulk of other subgroups, dark inclusion and matrix alone[33]) and different pressures from 100 to 2000 bars show similar outputs.

**Data availability**. With the exception of the thermodynamical properties of kirschsteinite calculated and fully described in the main text, other data sets used in the current study are referenced in the methods section. The original data in this article are available from the corresponding author on reasonable request.

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

## Acknowledgements

We thank Seth Jacobson and Nathalie Vigier for their corrections that greatly improved this article. Benoit Dubacq is warmly acknowledged for his invaluable support when calculating kirschsteinite thermodynamical data. We are grateful for the samples provided by the MNHN (Paris), the Statens Naturhistoriske Museum (Copenhagen) and the Smithsonian NMHN (Washington, DC). We thank Olivier Tottereau, Suzanne Jacomet and Valentina Batanova for technical assistance. This project was financially supported by Université de la Côte d'Azur (UCA) IDEX Académie 3, BQR from the Observatoire de la Côte d'Azur (OCA), Programme National de Planétologie (PNP) CNRS and CSI from the Université de Nice Sophia Antipolis. Géoazur and Lagrange are also acknowledged for their support.

## Author contributions

C.G and G.L. conceived and led this study, analyses and interpreted the results and wrote and corrected the manuscript.

## Additional information

**Competing financial interests:** The authors declare no competing financial interests.

