## [Peer Review File · Nature Communications]

Reviewers' comments:

Reviewer #1 (Remarks to the Author):

Dear Drs. Ganino and Libourel,

Thank you for the obvious care and effort that went into this work. The paper is well written such that it is enjoyable to read and the figures are clear and well described in captions. The thermodynamic calculations are sensible and their implications for the possible production of super-critical fluids represent a new model for formation of secondary phases in H₂O depleted geothermal environments across a temperature range of 300-600°C. Super critical fluids as drivers of secondary alteration are interesting and (although not discussed in the manuscript) potentially explain how pervasive fluid actions were possible in low permeability mediums without leaving evidence for elemental mobilizations. Forming secondary phases under reducing conditions at near to the iron magnetite redox buffer is also able to explain the curious associations of reduced silicates, magnetite ± serpentine in matrix of CV samples. A major strength of the work is an ability to relate all of the most abundant secondary products in CV matrices (e.g. andradite, hedenbergite, kirschsteinite, wollastonite, magnetite fayalite, serpentine) to a metamorphic trend, and an evolving environment with respect to temperature and aSiO₂. Importantly, the thermodynamic predictions of stable phase associations agree well with the described petrography of the CV samples from different groups (from your observations workers and in the literature). This links CV groups genetically, all formed under reduced conditions potentially at the same crustal depth, and I believe that this result allows for a simpler picture of the parent body asteroid than the "onion shell". The idea that all CV groups formed at a common level/depth in an asteroid is further consistent with the tendency for clasts of reduced CV to be found in oxidized CV. Forming all CVs at a common level removes the need for complex mixing of materials from different depth layers during putative brecciation and break up of parent bodies during impacts e.g. Elkins-Tanton et al., 2011 (for which CV petrography records little evidence). The work therefore bridges the gap between modeling and petrography in a consistent manner and presents conclusions that appeal from the perspective of Occam's razor. The inferred formation temperatures for these secondary phases under conditions of low aSiO agree well with those inferred from the maturity of organic matter in matrix and this is a significant first.

Therefore I think that this paper is an important contribution to the field that has implications for our understanding of the evolutionary history of CV chondrites and carbonaceous chondrites more generally.

I have annotated the submitted .pdf file with many additional comments, some significant and many minor. My major questions and other comments/suggestion not made above are also summarized below.

Main Questions

- I noted above that it is positive that the temperature ranges your modeling suggests agree well with the range inferred for organics. However, on the basis of organic maturity Allende is usually suggested to have experienced higher temperatures than Vigrano or Effremovka. Similarly on basis of phyllosilicates forming from enstatite in chondrules, it has been suggested Allende experienced higher temperatures than the other CV (Bonai). Your modeling suggests the opposite, that Allende records lower temperatures than the reduced CV. Can this be explained?
- pH of the fluid does not appear to be considered in the manuscript. If conditions are acidic serpentine formation is severely inhibited at any temperature range (e.g. Ohnishi and Tomeoka, 2007) is there a reason to exclude pH being acidic in the CV parent body and can this be mentioned? Perhaps worth noting Zolensky et al (1989: Icarus 78, 411-425) relate preservation of metal to alkaline conditions and CV do contain metal.

•How does the $\Delta 17\text{O}$ value allow for estimation of the initial water abundance in CV chondrites? I understand that differences in $\Delta 17\text{O}$ between samples can indicate different water/rock ratios but I'm not certain how the absolute value is used to infer a initial water abundance.

Main Comments/suggestions:

•I note that CK are suggested to be from the same stratified body as CV (Elkins-Tanton L. et al. 2011) and there are attempts to place CM and CO on same stratified parent body (Greenwood et al., 2014, LPSC abstract #2610). My point is that this model of vertically stacked asteroids with distinct physiochemical environments, varying with depth in the body, is fairly pervasive generally. This paper focuses on CV but it might be worth emphasizing the broader context.

•In CV phyllosilicates are most often observed on/around chondrules where saponite tends to be identified more than does serpentine. This preference for saponite over serpentine is something that has been related to a high activity of Silica from alteration of pyroxene and mesostases glasses in chondrules (e.g. Ohnishi and Tomeoka, *Meteoritics & Planetary Science* 42, 49–61 (2007)). Rather than contradict your model this might be interpreted as indicating very localized hydration only, perhaps confined to small areas where W/R were high enough to allow hydration reactions. The bulk of the asteroidal environment could still be reduced H₂O depleted setting as you argue (Perhaps water sits at contacts of chondrules and matrix).

•A feature of Darcy flow, for example in aquifer on Earth, is that the rate of flow varies different directions, such that different physiochemical environments may exist spatially and overlap in time. Meaning that in different regions of the same crustal level aSiO₂ and temperature could vary such that assemblages typical of CVOx and CVRed form contemporaneously. Further promoting heterogeneity in alteration and not requiring a stratified onion shell.

•Bulk modal mineralogy determined by PSD-XRD (Howard et al., 2010 – cited in the manuscript) and Mossbauer (Bland et al., (2000) *Meteorit. Planet. Sci.* 35(suppl), A28) both indicate that the abundance of magnetite is not consistent with the classification of CV as reduced or oxidized. For example magnetite abundances in reduced samples Vigrano 1.4%, and Effremovka 1.8% as compared to oxidized CVs Allende <1% and Mokoia 1.6%. This would seem consistent with forming all in an environment with relatively similar oxidation state, near to the I-M redox barrier.

•The data presented may preclude the requirement for a stratified crust as source of carbonaceous chondrite groups but do not contradict paleomagnetic evidence for differentiated asteroidal interiors, I think that this should be noted in the conclusions of the paper.

Minor Comments/suggestions:

•The supplementary table S1 is a valuable compilation of your observations and the literature. Subject to constraints on figure numbers it would be nice to include in the main text.

•Higher resolution X-ray maps would benefit the work.

Again, Thank you for this contribution.

Kieren Howard.

Reviewer #2 (Remarks to the Author):

General comments:

1. L16. ... overlooked low silica activity – Please note that the authors missed the paper entitled “In situ growth of Ca-rich rims around Allende dark inclusions” by Krot et al. (2000) *Geochemistry International* 38, S351-S368, where role of $a\text{SiO}_2$, $f\text{O}_2$ ($\text{H}_2/\text{H}_2\text{O}$ ratio), as well as other parameters ($a\text{Ca}^{2+}$, $a\text{Fe}^{2+}$, mole fractions of hedenbergite and fayalite in high-Ca pyroxene and olivine, respectively) in the formation of secondary Ca-Fe-rich secondary phases were discussed.

2. It is not clear how the authors constrained $a\text{SiO}_2$ in the CV fluid. Krot et al. (2000) used $a\text{SiO}_2$ estimated by Petaev and Mironenko for the aqueous solutions equilibrated with the Kaba matrix, which is very low ($< 10^{-3}$).

Petaev, M.I. and Mironenko, M.V., Thermodynamic modeling of aqueous alteration in CV chondrites, LPI Technical Report, 1997, no. 97-02, part I, pp. 49-50.

Note also that $a\text{SiO}_2$ in serpentinites are of the order of $10^{-2.5}$ to 10^{-5} .

3. In addition to $a\text{SiO}_2$ and $f\text{O}_2$, other intensive parameters need to be considered, such as $a\text{Ca}^{2+}$ and $a\text{Fe}^{2+}$, to define the stability fields of secondary Ca,Fe-rich silicates. Oxidation of Fe,Ni-metal nodules to magnetite followed by replacement of magnetite by hedenbergite, fayalite, and andradite, clearly indicate that Ca, Fe and Si were mobile elements during the alteration.

4. I am surprised by the lack of reference to a paper recently published in *Science Advance* by Y. Marrocci, M. Chaussidon, L. Piani, and G. Libourel, where it is concluded that magnetite in CV chondrules had a nebular, igneous origin. In contrast, in the current paper, C. Ganino and G. Libourel argue that alteration of CV parent body by a fluid was buffered at iron-magnetite redox conditions, suggesting that magnetite had an asteroidal origin.

5. The authors favor the existence of a dynamo on the CV parent body, which, if correct, requires igneous differentiation and formation of Fe,Ni-metal core. As a result, thermal structure of the CV chondritic crust is expected to be stratified, at least in terms of peak metamorphic temperature experienced by CV chondrites consistent with a range of petrologic subtypes (3.1 to >3.6) among CVs.

6. Iron-magnetite redox conditions can hardly be called reduced compared to solar nebula conditions.

7. The authors have not addressed the lack of magnetite in the reduced CV chondrites Efremovka and Leoville. This was the primary reason to divide CVs into the reduced and oxidized subgroups (McSween, 1977).

Minor comments:

L34 – proceeds

L33-35. Formation of secondary minerals in an asteroidal setting is inferred from mineralogical observations, thermodynamic analysis, and oxygen-isotope compositions (see Brearley and Krot, 2012 for a review). ^{53}Mn - ^{53}Cr ages just added another piece of evidence.

L36 - ... unprocessed crust – Paleomagnetic records were identified in secondary minerals. Therefore, the crust cannot be unprocessed, it experienced alteration to various degrees under different conditions.

L37 – ^{26}Al (refs 6, 7)

L57 – not exactly. Weisberg et al. used also bulk oxygen isotopic compositions and chondrule/matrix ratio. In addition, Weisberg et al. favored alteration of CV chondrites in the solar nebula.

L68 – What about magnetite in the reduced CV chondrites Efremovka and Leoville?

L82 – see general comments 1–3.

L102 – Pressure of 2000 bars is too high for a crust of an asteroidal body. Tensile strength of a rock is ~ 100 bar.

L102 – How did you estimate a plausible range of $a\text{SiO}_2$ values?

L130. In the presence of aqueous solutions, pure fayalite is unstable above 300°C , more magnesian olivine is stable instead (see Zolotov et al., 2006). In the Bali-like oxidized CVs, pure fayalite is often overgrown by more magnesian olivine indicating temperature increase during alteration, contrary to the path indicated in Figure 4.

Zolotov M.Yu., Mironenko M.V., and Shock E.L. (2006) Thermodynamic constraints on fayalite formation on parent bodies of chondrites. *Meteorit. Planet. Sci.* 41, 1775–1796.

L128 – Formation of kirschsteinite at high temperature ($>450^\circ\text{C}$) inferred in the paper is different from the formation temperature at $\sim 100^\circ\text{C}$ estimated by Krot et al. (2000).

L129 – The inferred formation conditions of fayalite ($a\text{SiO}_2 > 10^{-1}$) could be unrealistic for CV alteration (see general comment #2).

L148 – andradite is a common mineral in rims around Allende dark inclusions formed in situ, which experienced higher temperature alteration than the reduced and Bali-like oxidized CVs.

It seems unlikely that redox conditions stayed constant during CV alteration, e.g., serpentinization reactions produce significant amount of H_2 resulting in decrease of $f\text{O}_2$. Sudden loss of H_2 through cracks will result in increase of $f\text{O}_2$ (Zolotov et al., 2006). Oxidation of metal to magnetite and replacement of magnetite by fayalite are also consistent with variable redox conditions during CV alteration.

L178. Hydrous phases are common in the Bali-like CVs.

L190. Please address the lack on magnetite in the reduced CVs.

L197 – This is the first time where $a\text{CaO}$ and $a\text{FeO}$ mentioned, but no analysis of these parameters on the stability fields of secondary phases was done (see Krot et al., 2000).

L203-205. Where do these estimates come from?

L204. Kaba contains abundant phyllosilicates and hedenbergite. It is unlikely that it experienced high temperature ($\sim 500^\circ\text{C}$) metasomatic alteration. This conclusion is also inconsistent with the text on L228-230.

L206 – This is not a novel conclusion. Similar conclusion was reached by Krot et al. in many papers (e.g., 1998, 2000).

L215-217 – For estimates of silica activity in CV fluid see Petaev, M.I. and Mironenko, M.V., Thermodynamic modeling of aqueous alteration in CV chondrites, LPI Technical Report, 1997, no. 97-02, part I, pp. 49-50.

L239-244 – Fayalite always coexists with phyllosilicates in meteorites of very low petrologic types, Kaba (CV3.1), Semarkona (LL3.0), and ALH 77307 (CO3.0). It is unclear how subsolidus high-temperature crystallization of fayalite can explain the lack of corrosion of fayalite by the

surrounding phyllosilicates, which are hypothesized to have formed during retrograde metamorphism. In addition, Semarkona and ALH 77307 contain abundant amorphous ferromagnesian silicates in their matrices which could not survive during subsolidus high-temperature crystallization of fayalite.

L250-254. This is not the original idea. The similar conclusion was reached by Krot et al. based on real O-isotopic compositions of secondary minerals, including hedenbergite, andradite, fayalite, magnetite etc. measured in situ by SIMS. The latest results were reported at the Meteoritical Society meeting in Berlin.

Reviewer #3 (Remarks to the Author):

This is a beautiful piece of work. The central idea is elegant, with profound implications for our understanding of the geological history of primitive asteroids. Regarding CV3 chondrites in particular, these are the meteorites that contain the oldest dated materials, defining the age of the solar system. A radical re-interpretation of the geological history of the CV3 parent body therefore has significant implications. Like all great ideas, it seems obvious in retrospect. The authors insight is based on traditional petrography and thermodynamics. The fact that they have applied this to some of the most studied rocks on Earth, and seen something that all other workers have missed, speaks to its originality. It should be published in Nature Communications (although the text requires significant attention – see below).

SPECIFIC COMMENTS AND QUESTIONS FOR THE AUTHORS

How does their model map onto other estimates of temperature e.g. Raman? The authors note that the range in temperature fits with other estimates for the CV3s, but I believe that all other indicators have reversed the metamorphic sequence set out in this paper - putting Allende at significantly higher T than Efremovka, Vigarano, or Leoville. This seems to be supported by the variable preservation of primordial components in these rocks. The authors don't speak to this in the paper. It should be discussed.

A related question is grain size. The reduced CVs, and Kaba, have much smaller grain size than Allende. How do we minimise recrystallisation at these temperatures? Returning to the point outlined above, why is Allende matrix coarser grained than these other rocks, if they experienced higher temperatures than Allende?

The authors state (235-237) that their model '...provides an alternative to the classical aqueous alteration models proposed for the CV3 chondrite parent body, in which Ca-Fe-rich pyroxenes... are inferred to form at relatively low temperatures...'. On the contrary, there is evidence that they formed at very high T – much higher in fact than the estimates here. TEM studies of pyroxene polymorphs in Allende matrix have provided strong evidence for a high temperature origin (>1300K) followed by very rapid cooling [Brenker et al. 2000; Brenker and Krot 2004].

Finally, implications. One of the most significant implications – '...short periods of hydrothermal activity on the parent body (<10⁴-10⁵ yr...' – is not developed. It should not be for the reader to wonder what the significance of this is. Explore it against the background of existing models – e.g. the hypothesis that the CV3 parent body is differentiated. Similarly, the statement that these results '...collectively indicate that fluid-assisted metamorphism on CV3 parent asteroid should not be considered as a continuous, protracted event, but rather as the cumulative effects of hydrothermal activity changes.' What does this mean? If hydrothermal activity is not continuous, what is it? What is the mechanism that drivers discontinuous hydrothermal alteration? Please discuss.

THE TEXT

The text needs a very substantial re-write, for clarity, typos, and grammar. It does not seem to have had more than a cursory read-through. I'll not make corrections throughout – that's the author's job – but I'll use the abstract as an example. Regarding clarity, the sentence 'The various

lithologies in CV3 chondrites are thus inferred to be fragments of one heterogeneously silica undersaturated fluid percolated asteroid via porous flow' is pretty impenetrable, and may be grammatically incorrect – although that depends on its meaning. At minimum, it needs clarifying. The abstract should be understandable to a general reader. If its not intelligible to a specialist then there is a problem. There are numerous typos. Again, taking the abstract as an example (e.g. should be 'close' rather than 'closed' in '...closed to the iron-magnetite redox buffer...'). And tense (past / present etc) should be consistent throughout the MS: '...secondary alteration phases formed at relatively low temperatures...', '...CV chondrites witness different physicochemical...'.

REVIEWER 1

Main Questions

1- I noted above that it is positive that the temperature ranges your modeling suggests agree well with the range inferred for organics. However, on the basis of organic maturity Allende is usually suggested to have experienced higher temperatures than Vigrano or Effremovka. Similarly on basis of phyllosilicates forming from enstatite in chondrules, it has been suggested Allende experienced higher temperatures than the other CV (Bonal). Your modeling suggests the opposite, that Allende records lower temperatures than the reduced CV. Can this be explained?

→ *That other phases (e.g., organic matter, Bonal et al., 2016) or phase assemblages (e.g., calcic amphibole, anthophyllite and talc as alteration products of primary chondrule glass in Allende, Brearley, 1997) suggest higher (or different) temperatures than our assessments from Ca-Fe-rich secondary assemblage stability fields is not inconsistent with our model.*

One of the main findings of this work is to show that the stability of secondary phases in CV chondrites are controlled by a complex set (multivariant space) of intensive variables, including P, T and the chemical potentials of the system. In this frame, we have shown that the silica activity, a_{SiO_2} and the fugacity of oxygen, f_{O_2} are playing a key role. Due to the interplay of these parameters, this means that a phase or a phase assemblage could be stable over a range of these parameters. As shown in Fig. 3 as an example, hydrous secondary phases can be stable at low temperature for a low or very silica activity of the system ($\log a_{\text{SiO}_2} \ll -1$), or in contrast can occur at relatively high temperature for a higher silica activity.

While beyond the scope of this paper devoted to secondary phases in CV matrices, the interplay of these parameters explains well why anthophyllite and talc (could have) formed in chondrule mesostasis at a higher temperature (Brearley, 1997) than in matrices due to the ubiquitous occurrence of olivine and low-Ca pyroxene that buffer during the “alteration” of the chondrule mesostasis the local silica activity according to $\text{Mg}_2\text{SiO}_4 (\text{Fo}) + \text{SiO}_2 (\text{Qtz}) = \text{Mg}_2\text{Si}_2\text{O}_6 (\text{En})$ chondrule (see Fig. 9 below in Klein et al., GCA, 2009).

Fig. 9. Temperature-SiO₂ activity plot depicting the phase relations in the system MgO-SiO₂-H₂O. Note the pressure is 200 MPa

In matrices on the other hand, which is overwhelmingly dominated by (ferroan) olivines, the silica activity is not buffered and is simply controlled by the composition of the fluid phase and/or its level of interaction with the silica activity un-buffered matrix. In this case and as shown in our Fig. 3 for the

FeO-SiO₂-H₂O system or in the above figure for the MgO-SiO₂-H₂O system, hydrous phases (e.g., saponite or serpentine-like) could crystallize but only at a well lower temperature. From this example it is thus clear that different phases or phases assemblages may record different temperatures of formation, as on Earth in the case of a prograde or retrograde metamorphism. The same can be said about the evolution of the organic matter in the T-fO₂ space.

In addition, the different Ca-Fe rich phases assemblages found in a small area (few 10's of microns or veinlets, see fig. 1) in CV matrices suggest that they results from the heterogeneous percolation of fluid(s) (Darcy flow) and its(their) localized interaction(s) with the surrounding matrix. Depending on the composition of the fluid, its temperature and the local mineralogy of the matrix on the way, it seems then unavoidable that the resulting secondary phases assemblages must be very diverse and record different P, T and chemical activities. And as in a case of an inverse problem, these different phases assemblages can then allow us to track the conditions reigning in the parent body.

2- pH of the fluid does not appear to be considered in the manuscript. If conditions are acidic serpentine formation is severely inhibited at any temperature range (e.g. Ohnishi and Tomeoka, 2007) is there a reason to exclude pH being acidic in the CV parent body and can this be mentioned? Perhaps worth noting Zolensky et al (1989: Icarus 78, 411–425) relate preservation of metal to alkaline conditions and CV do contain metal.

→ This idea is interesting but we notice that the decimal logarithm of the reciprocal of the hydrogen ion activity, aH^+ , in a solution is used for aqueous solution or liquid aqueous condensate form a gas phase, but not for gases or supercritical fluid, in which nature of the metal-ligand complexes, the metal speciation and solubility are related to the water partial pressure.

3- How does the $\Delta 17O$ value allow for estimation of the initial water abundance in CV chondrites? I understand that differences in $\Delta 17O$ between samples can indicate different water/rock ratios but I'm not certain how the absolute value is used to infer a initial water abundance.

→ Agreed. We modified the manuscript following this suggestion.

Main Comments/suggestions:

4- I note that CK are suggested to be from the same stratified body as CV (Elkins-Tanton L. et al. 2011) and there are attempts to place CM and CO on same stratified parent body (Greenwood et al., 2014, LPSC abstract #2610). My point is that this model of vertically stacked asteroids with distinct physiochemical environments, varying with depth in the body, is fairly pervasive generally. This paper focuses on CV but it might be worth emphasizing the broader context.

→ In the present state of our knowledge of the secondary phase assemblages of the other chondrite groups it would be too speculative to broaden our model of CV crust to other chondrite parent bodies. However, such a detailed survey of secondary phase assemblages in CK, CO or CM could be of major interest to test their consistency with our thermodynamic analyses and its implications of heterogeneously fluid percolated body instead of stratified crust. If well-more oxidized (FMQ+2 to FMQ+4.5, Righter and Neff, 2007) redox conditions in CK seems to preclude formation in a same environment, it is of note that assemblage Kst+Fa+Hd in CO chondrites (Doyle et al. 2014, figure 1c) can be interpreted in the frame of our thermodynamic analyses and indicating high silica activity and high temperature (>800°C). Whereas such assemblage are not described as far in CV, to a certain extent, this feature could be consistent with very high temperature documented by Brenker et al. (2000)

5- In CV phyllosilicates are most often observed on/around chondrules where saponite tends to be identified more than does serpentine. This preference for saponite over serpentine is something that has been related to a high activity of Silica from alteration of pyroxene and mesostases glasses in chondrules (e.g. Ohnishi and Tomeoka, Meteoritics & Planetary Science 42, 49–61 (2007)). Rather than contradict your model this might be interpreted as indicating very localized hydration only, perhaps confined to

small areas where W/R were high enough to allow hydration reactions. The bulk of the asteroidal environment could still be reduced H₂O depleted setting as you argue (Perhaps water sits at contacts of chondrules and matrix).

→ Thank you for this remark. This item is also addressed in the response of point 1.

6- A feature of Darcy flow, for example in aquifer on Earth, is that the rate of flow varies different directions, such that different physiochemical environments may exist spatially and overlap in time. Meaning that in different regions of the same crustal level aSiO₂ and temperature could vary such that assemblages typical of CVOx and CVRed form contemporaneously. Further promoting heterogeneity in alteration and not requiring a stratified onion shell.

→ Agreed. This is the main idea of this manuscript, but we include the wording of this reviewer in the text in order to clarify the implication of a Darcy flow process.

7- Bulk modal mineralogy determined by PSD-XRD (Howard et al., 2010 – cited in the manuscript) and Mossbauer (Bland et al., (2000) Meteorit. Planet. Sci. 35(suppl), A28) both indicate that the abundance of magnetite is not consistent with the classification of CV as reduced or oxidized. For example magnetite abundances in reduced samples Vigrano 1.4%, and Effremovka 1.8% as compared to oxidized CVs Allende <1% and Mokoia 1.6%. This would seem consistent with forming all in an environment with relatively similar oxidation state, near to the I-M redox barrier.

→ Agreed. Our survey indeed confirms occurrence of magnetite in the matrix of reduced CV chondrites. This observation was one of the starting point of our questioning concerning the classification of CV as reduced or oxidized. These two references were added in the manuscript.

8- The data presented may preclude the requirement for a stratified crust as source of carbonaceous chondrite groups but do not contradict paleomagnetic evidence for differentiated asteroidal interiors, I think that this should be noted in the conclusions of the paper.

→ Agreed. This has been added and clarified in the manuscript.

Minor Comments/suggestions:

9- The supplementary table S1 is a valuable compilation of your observations and the literature. Subject to constraints on figure numbers it would be nice to include in the main text.

→ Agreed. It will be proposed to the editor.

10- Higher resolution X-ray maps would benefit the work.

→ The images were low-res in the pdf compiled for the review, but high-resolution images will be submitted if the publication is accepted. Ultra-High resolution X-ray maps are also available and will be proposed as supplementary files 4.

REVIEWER 2

Reviewer #2 (Remarks to the Author):

General comments:

1. L16. ... overlooked low silica activity – Please note that the authors missed the paper entitled “In situ growth of Ca-rich rims around Allende dark inclusions” by Krot et al. (2000) *Geochemistry International* 38, S351-S368, where role of a_{SiO_2} , f_{O_2} ($\text{H}_2/\text{H}_2\text{O}$ ratio), as well as other parameters ($a_{\text{Ca}^{2+}}$, $a_{\text{Fe}^{2+}}$, mole fractions of hedenbergite and fayalite in high-Ca pyroxene and olivine, respectively) in the formation of secondary Ca-Fe-rich secondary phases were discussed.

→ *This article was not cited because its results and implications have been summarized recently in Brearley and Krot (2012) and mentioned in Krot et al. 1998, 2004, we already cited. However, due the specific point raised by the reviewer, this article is now quoted. Nevertheless there is a fundamental difference between our interpretations for the formation of the same Ca-Fe-rich phases. In Krot et al. (1998, 2000, 2004, all cited in our article) they assumed that these phases precipitated from aqueous solution at low temperature (<250°C), whereas we interpret here the Ca-Fe-rich phase assemblages as the result of the interaction of the matrix with supercritical H_2O -poor fluid at higher temperature (210–610°C). For them a_{SiO_2} , $a_{\text{Ca}^{2+}}$, $a_{\text{Fe}^{2+}}$ are “effective” concentration of these elements in the aqueous fluid whereas in our case, it corresponds to the effective concentration of SiO_2 in the bulk system, including the supercritical fluid or vapor and the percolated matrix.*

2. It is not clear how the authors constrained a_{SiO_2} in the CV fluid. Krot et al. (2000) used a_{SiO_2} estimated by Petaev and Mironenko for the aqueous solutions equilibrated with the Kaba matrix, which is very low (< 10⁻³).

Petaev, M.I. and Mironenko, M.V., Thermodynamic modeling of aqueous alteration in CV chondrites, LPI Technical Report, 1997, no. 97-02, part I, pp. 49-50.

Note also that a_{SiO_2} in serpentinites are of the order of 10^{-2.5} to 10⁻⁵.

→ *a_{SiO_2} is not calculated and is a free parameter in our thermodynamic phase diagram. Note also that our a_{SiO_2} is not that of an aqueous solution, but the one of the bulk system (see above). That a_{SiO_2} in CV matrix (e.g., Allende) is highly variable is attested by the occurrence of rankinite, larnite, (very low), nepheline/sodalite (intermediate) and talc and anthophyllite (relatively high), and compatible with those inferred for serpentinites (see Fig. 9 above in Klein et al., GCA, 2009)*

3. In addition to a_{SiO_2} and f_{O_2} , other intensive parameters need to be considered, such as $a_{\text{Ca}^{2+}}$ and $a_{\text{Fe}^{2+}}$, to define the stability fields of secondary Ca,Fe-rich silicates. Oxidation of Fe,Ni-metal nodules to magnetite followed by replacement of magnetite by hedenbergite, fayalite, and andradite, clearly indicate that Ca, Fe and Si were mobile elements during the alteration.

→ *Agreed. $a_{\text{Ca}^{2+}}$ and Fe^{2+} are not considered here because these are ionic species compatible only with low temperature aqueous chemistry.*

However the role of a_{CaO} and a_{FeO} have been explored. The role of a_{CaO} in the system is more or less the opposite of the one of a_{SiO_2} because of the stoichiometry of the phase of interest. Ca-rich phases are also silica-poor phases.

As outlined by the reviewer (his second sentence) Fe needs to be a mobile element too, since it is the other major element of the phases we are looking at. Its role is more complex since several valences could be involved in the present redox conditions. High a_{FeO} clearly promotes the crystallization of fayalite, as it is suggested by the close association between the oxidation of Fe,Ni metal, magnetite and fayalite ± hedenbergite.

4. I am surprised by the lack of reference to a paper recently published in Science Advance by Y. Marrocci, M. Chaussidon, L. Piani, and G. Libourel, where it is concluded that magnetite in CV

chondrules had a nebular, igneous origin. In contrast, in the current paper, C. Ganino and G. Libourel argue that alteration of CV parent body by a fluid was buffered at iron-magnetite redox conditions, suggesting that magnetite had an asteroidal origin.

→ The article cited here focused on the formation of magnetite in the chondrule rather than in the matrix. Owing to the new results obtained for this study, we don't agree with the interpretation proposed by Marrocchi et al. that magnetite in CV chondrules had a nebular, igneous origin. We now cite this reference and underline our disagreement.

5. The authors favor the existence of a dynamo on the CV parent body, which, if correct, requires igneous differentiation and formation of Fe,Ni-metal core. As a result, thermal structure of the CV chondritic crust is expected to be stratified, at least in terms of peak metamorphic temperature experienced by CV chondrites consistent with a range of petrologic subtypes (3.1 to >3.6) among CVs.

→ The reviewer point is correct at a large scale and if the heat transfer is mainly conductive. However, as shown by Fu et Elkins-Tanton, EPSL, 2014, the progressive heating of the chondritic protolith will result in extensive devolatilization and the formation of a extensive network of percolation route, the heat being transfer (advection/convection) by the heterogeneous Darcy flow. In this case, no stratification of the crust is expected.

6. Iron-magnetite redox conditions can hardly be called reduced compared to solar nebula conditions.

→ Agreed. However because we are discussing "metamorphic" processes on an asteroidal body, we have taken the redox reference with respect to the Fayalite-Magnetite-Quartz (FMQ) buffer as on Earth. Iron-Wustite oxygen buffer conditions are therefore more reduced by 2-3 order of magnitude. In the text modifications have been done to specify that the reducing character of IM is with respect to FMQ.

7. The authors have not addressed the lack of magnetite in the reduced CV chondrites Efremovka and Leoville. This was the primary reason to divide CVs into the reduced and oxidized subgroups (McSween, 1977).

→ Our observation does not support the McSween's classification, as well as those of the reviewer 1: " Bulk modal mineralogy determined by PSD-XRD (Howard et al., 2010 – cited in the manuscript) and Mossbauer (Bland et al., (2000) Meteorit. Planet. Sci. 35(suppl), A28) both indicate that the abundance of magnetite is not consistent with the classification of CV as reduced or oxidized. For example magnetite abundances in reduced samples Vigrano 1.4%, and Effremovka 1.8% as compared to oxidized CVs Allende <1% and Mokoia 1.6%. This would seem consistent with forming all in an environment with relatively similar oxidation state, near to the IM redox barrier." This argument is now in the manuscript.

Minor comments:

L34 – proceeds → corrected

L33-35. Formation of secondary minerals in an asteroidal setting is inferred from mineralogical observations, thermodynamic analysis, and oxygen-isotope compositions (see Brearley and Krot, 2012 for a review). ⁵³Mn-⁵³Cr ages just added another piece of evidence.

→ Corrected

L36 - ... unprocessed crust – Paleomagnetic records were identified in secondary minerals. Therefore, the crust cannot be unprocessed, it experienced alteration to various degrees under different conditions.

→ Modified to "unmelted"

L37 – 26Al (refs 6, 7)

→corrected

L57 – not exactly. Weisberg et al. used also bulk oxygen isotopic compositions and chondrule/matrix ratio. In addition, Weisberg et al. favored alteration of CV chondrites in the solar nebula.

→modified

L68 – What about magnetite in the reduced CV chondrites Efremovka and Leoville?

→ See the reply to your general comment 7: the abundance of magnetite is not consistent with the classification of CV as reduced or oxidized.

L82 – see general comments 1–3.

L102 – Pressure of 2000 bars is too high for a crust of an asteroidal body. Tensile strength of a rock is ~ 100 bar.

→ This is already discussed “Computations using bulk composition of other CV3 lithologies (including bulk of other subgroups, dark inclusion and matrix alone) : the thermodynamic data are generally published (and best-constrained) at 2000 bars, and different pressures from 100-2000 bars show similar outputs.” See below (200 bars (left) versus 2000 bars (right)).

L102 – How did you estimate a plausible range of aSiO2 values?

→ See reply to your general comment 2: aSiO₂ is not calculated and is a free parameter but in order to have rankinite, larnite, kirschsteinite system must have very low or low aSiO₂.

L130. In the presence of aqueous solutions, pure fayalite is unstable above 300°C, more magnesian olivine is stable instead (see Zolotov et al., 2006). In the Bali-like oxidized CVs, pure fayalite is often overgrown by more magnesian olivine indicating temperature increase during alteration, contrary to the path indicated in Figure 4.

Zolotov M.Yu., Mironenko M.V., and Shock E.L. (2006) Thermodynamic constraints on fayalite formation on parent bodies of chondrites. Meteorit. Planet. Sci. 41, 1775–1796.

→ This refers to stability in the presence of aqueous solutions. If other conditions are invoked then fayalite can be stable above 300°C

L128 – Formation of kirschsteinite at high temperature (>450°C) inferred in the paper is different from the formation temperature at ~100°C estimated by Krot et al. (2000).

→ *We clarify this point. Our model of formation of kirschsteinite is different and do not consider precipitation from an aqueous fluid.*

L129 – The inferred formation conditions of fayalite ($a_{\text{SiO}_2} > 10^{-1}$) could be unrealistic for CV alteration (see general comment #2).

→ *see reply to your general comment 1 and how we have calculated a_{SiO_2} : the bulk effective concentration of SiO_2 in the bulk system (including the supercritical fluid or vapour and the percolated matrix). As shown by fig. 3 and 4, Fayalite can form only at relatively high a_{SiO_2} , consistent with the FMQ reaction.*

L148 – andradite is a common mineral in rims around Allende dark inclusions formed in situ, which experienced higher temperature alteration than the reduced and Bali-like oxidized CVs.

→ *Agreed, nevertheless andradite is abundant in Bali-like CVs as well as in Kaba-Mokoia and Efremovk, while less frequent. We did not observe any difference in the type of occurrence of andradite in the different subgroups.*

It seems unlikely that redox conditions stayed constant during CV alteration, e.g., serpentinization reactions produce significant amount of H_2 resulting in decrease of f_{O_2} . Sudden loss of H_2 through cracks will result in increase of f_{O_2} (Zolotov et al., 2006). Oxidation of metal to magnetite and replacement of magnetite by fayalite are also consistent with variable redox conditions during CV alteration.

→ *Contrary to what would occurred in an aqueous dominated chemical system, where the fluid composition can be highly variable in term of redox conditions, here we consider that the oxygen fugacity is buffered by the matrix composition. If changes of oxygen fugacities close to the IM-redox conditions are possible of course, it seems for us that the formation of fayalite is more sensitive to changes in the silica activity ($f_{\text{a}} = \text{iron} + \text{O}_2 + \text{SiO}_2$, $f_{\text{a}} = \text{mt} + \text{SiO}_2$).*

L178. Hydrous phases are common in the Bali-like CVs.

→ *Agreed, these phases are present but in low abundance (3.5 to 4.2% in CVOxB, Howard et al. 2010).*

L190. Please address the lack on magnetite in the reduced CVs.

→ *This is now addressed citing Howard et al. (2010) and Bland et al. (2000)*

L197 – This is the first time where a_{CaO} and a_{FeO} mentioned, but no analysis of these parameters on the stability fields of secondary phases was done (see Krot et al., 2000).

→ *see response to major comment n°3*

L203-205. Where do these estimates come from?

→ *$\text{H}_2/\text{H}_2\text{O}$, $\text{H}_2/\text{H}_2\text{S}$, and CO/CO_2 are estimated for the fixed temperature of 500°C and the IM-redox buffer using the Ellingham's (1944) diagram. It is now added in methods. Ellingham, H. J. T. (1944). Transactions and communications. J. Soc. Chem. Ind.(London), 63(5), 125.*

L204. Kaba contains abundant phyllosilicates and hedenbergite. It is unlikely that it experienced high temperature (~500°C) metasomatic alteration. This conclusion is also inconsistent with the text on L228-230.

→ *see response 1 reviewer 1*

L206 – This is not a novel conclusion. Similar conclusion was reached by Krot et al. in many papers (e.g.,

1998, 2000).

→ Agreed. We modified the manuscript accordingly.

L215-217 – For estimates of silica activity in CV fluid see Petaev, M.I. and Mironenko, M.V., Thermodynamic modeling of aqueous alteration in CV chondrites, LPI Technical Report, 1997, no. 97-02, part I, pp. 49-50.

→ See general comment 2

L239-244 – Fayalite always coexists with phyllosilicates in meteorites of very low petrologic types, Kaba (CV3.1), Semarkona (LL3.0), and ALH 77307 (CO3.0). It is unclear how subsolidus high-temperature crystallization of fayalite can explain the lack of corrosion of fayalite by the surrounding phyllosilicates, which are hypothesized to have formed during retrograde metamorphism. In addition, Semarkona and ALH 77307 contain abundant amorphous ferromagnesian silicates in their matrices which could not survive during subsolidus high-temperature crystallization of fayalite.

→ Darcy flow percolation of the matrix allows heterogeneous alteration at the small scale therefore preserving high T and low T assemblages. We notice that corrosion is lacking for Fa50 as well.

L250-254. This is not the original idea. The similar conclusion was reached by Krot et al. based on real O-isotopic compositions of secondary minerals, including hedenbergite, andradite, fayalite, magnetite etc. measured in situ by SIMS. The latest results were reported at the Meteoritical Society meeting in Berlin.

→ The reference was not known when this article was submitted. It is now cited.

REVIEWER 3

Reviewer #3 (Remarks to the Author):

This is a beautiful piece of work. The central idea is elegant, with profound implications for our understanding of the geological history of primitive asteroids. Regarding CV3 chondrites in particular, these are the meteorites that contain the oldest dated materials, defining the age of the solar system. A radical re-interpretation of the geological history of the CV3 parent body therefore has significant implications. Like all great ideas, it seems obvious in retrospect. The authors insight is based on traditional petrography and thermodynamics. The fact that they have applied this to some of the most studied rocks on Earth, and seen something that all other workers have missed, speaks to its originality. It should be published in Nature Communications (although the text requires significant attention – see below).

SPECIFIC COMMENTS AND QUESTIONS FOR THE AUTHORS

1- How does their model map onto other estimates of temperature e.g. Raman? The authors note that the range in temperature fits with other estimates for the CV3s, but I believe that all other indicators have reversed the metamorphic sequence set out in this paper - putting Allende at significantly higher T than Efremovka, Vigarano, or Leoville.

→ *We reproduce here the same response to reviewer 1 because it is essentially the same question:*

That other phases (e.g., organic matter, Bonal et al., 2016) or phase assemblages (e.g., calcic amphibole, anthophyllite and talc as alteration products of primary chondrule glass in Allende, Brearley, 1997) suggest higher (or different) temperatures than our assessments from Ca-Fe-rich secondary assemblage stability fields is not inconsistent with our model.

One of the main finding of this work is to show that the stability of secondary phases in CV chondrites are controlled by a complex set (multivariant space) of intensive variables, including P, T and the chemical potentials of the system. In this frame, we have shown that the silica activity, a_{SiO_2} and the fugacity of oxygen, f_{O_2} are playing a key role. Due to the interplay of these parameters, this means that a phase or a phase assemblage could be stable over a range of these parameters. As shown in Fig. 3 as an example, hydrous secondary phases can be stable at low temperature for a low or very silica activity of the system ($\log a_{\text{SiO}_2} \ll -1$), or in contrast can occur at relatively high temperature for a higher silica activity.

While beyond the scope of this paper devoted to secondary phases in CV matrices, the interplay of these parameters explains well why anthophyllite and talc (could have) formed in chondrule mesostasis at a higher temperature (Brearley, 1997) than in matrices due to the ubiquitous occurrence of olivine and low-Ca pyroxene that buffer during the “alteration” of the chondrule mesostasis the local silica activity according to $\text{Mg}_2\text{SiO}_4(\text{Fo}) + \text{SiO}_2(\text{Qtz}) = \text{Mg}_2\text{Si}_2\text{O}_6(\text{En})$ chondrule (see Fig. 9 below in Klein et al., GCA, 2009).

Fig. 9. Temperature-SiO₂ activity plot depicting the phase relations in the system MgO-SiO₂-H₂O. Note the pressure is 200 MPa

In matrices on the other hand, which is overwhelmingly dominated by (ferroan) olivines, the silica activity is not buffered and is simply controlled by the composition of the fluid phase and/or its level of interaction with the silica activity un-buffered matrix. In this case and as shown in our Fig. 3 for the FeO-SiO₂-H₂O system or in the above figure for the MgO-SiO₂-H₂O system, hydrous phases (e.g., saponite or serpentine-like) could crystallize but only at a lower temperature. From this example it is thus clear that different phases or phases assemblages may record different temperatures of formation, as on Earth in the case of a prograde or retrograde metamorphism. The same can be said about the evolution of the organic matter in the T-fO₂ space.

In addition, the different Ca-Fe rich phases assemblages found in a small area (few 10's of microns or veinlets, see fig. 1) in CV matrices suggest that they result from the heterogeneous percolation of fluid(s) (Darcy flow) and its(their) localized interaction(s) with the surrounding matrix. Depending on the composition of the fluid, its temperature and the local mineralogy of the matrix on the way, it seems then unavoidable that the resulting secondary phases assemblages must be very diverse and record different P, T and chemical activities. And as in a case of an inverse problem, these different phases assemblages can then allow us to track the conditions reigning in the parent body.

2- This seems to be supported by the variable preservation of primordial components in these rocks. The authors don't speak to this in the paper. It should be discussed.

→Our article is limited to the study of secondary phases in the matrix. Nevertheless, that chondrules and CAIs experienced different degrees of alteration is already mentioned in different papers (e.g., Brearley and Krot, 2012). The reason why we didn't want to extend our study to the "primordial components" of the CV chondrites is mainly linked to the fact their composition and mineralogy is highly variable from one component to the other, and are significantly different from that of the matrix. Studying secondary phase stability fields in both matrix and in chondrules and CAI would have forced us to take into account the effect of the chemical composition which is by far one of the most complex and least constrained parameter. Here our strategy was simply to focus our study on the matrices to deal only with (almost) isochemical systems.

3- A related question is grain size. The reduced CVs, and Kaba, have much smaller grain size than Allende. How do we minimise recrystallisation at these temperatures?

→ This is an interesting point. However, we believe that the grain size of the CV chondrite matrix, mainly those of the ferroan olivines ($\approx Fa50$), is not related to the conditions inferred in this paper for the formation of the Ca-Fe-rich secondary phases. It is very frequent that veinlets formed by these secondary phases cross-cut the CV matrices, whatever their grain size.

4- Returning to the point outlined above, why is Allende matrix coarser grained than these other rocks, if they experienced higher temperatures than Allende?

→ See response above.

5- The authors state (235-237) that their model ‘...provides an alternative to the classical aqueous alteration models proposed for the CV3 chondrite parent body, in which Ca-Fe-rich pyroxenes... are inferred to form at relatively low temperatures...’. On the contrary, there is evidence that they formed at very high T – much higher in fact than the estimates here. TEM studies of pyroxene polymorphs in Allende matrix have provided strong evidence for a high temperature origin ($>1300K$) followed by very rapid cooling [Brenker et al. 2000; Brenker and Krot 2004].

→ We totally rewrote the section on the “compatibility of our result with other observables in CVs”. Our results document an “intermediate” temperature range that appear to match with the large majority of secondary phases. However, the presented results don’t rule out the possibility of locally higher or lower temperature. As explained in the response to major comment 1, the heterogeneous percolation of the fluid in the CV matrix (Darcy flow) is supposed to result in local and contrasted thermal conditions.

6- Finally, implications. One of the most significant implications – ‘...short periods of hydrothermal activity on the parent body ($<10^4-10^5$ yr...’ – is not developed. It should not be for the reader to wonder what the significance of this is.

→ The significance of this sentence is now explained. We add in the text: “Darcy flow would follow indeed different pathways through the crust permeability and different directions, explaining that different physicochemical environments both in space and time.”

7- Explore it against the background of existing models – e.g. the hypothesis that the CV3 parent body is differentiated.

→ Here we don’t question the fact that the CV3 parent body is differentiated. In the light of our results we simply suggest that the stratification of the upper crust of CV3 parent body as presented by Elkins Tanton et al., 2011 is not consistent with our results.

8- Similarly, the statement that these results ‘...collectively indicate that fluid-assisted metamorphism on CV3 parent asteroid should not be considered as a continuous, protracted event, but rather as the cumulative effects of hydrothermal activity changes.’ What does this mean? If hydrothermal activity is not continuous, what is it? What is the mechanism that drivers discontinuous hydrothermal alteration? Please discuss.

→ → Once again, we improved our description of the Darcy flow percolation and underlined the large variability we can expect in space and time. Even if the hydrothermal activity could be continuous, the pathways for Darcy flow move in space with time, resulting in a discontinuous hydrothermal alteration.

THE TEXT

The text needs a very substantial re-write, for clarity, typos, and grammar. It does not seem to have had more than a cursory read-through. I’ll not make corrections throughout – that’s the author’s job – but I’ll use the abstract as an example. Regarding clarity, the sentence ‘The various lithologies in CV3 chondrites are thus inferred to be fragments of one heterogeneously silica undersaturated fluid percolated asteroid via porous flow’ is pretty impenetrable, and may be grammatically incorrect – although that depends on its

meaning. At minimum, it needs clarifying.

9- The abstract should be understandable to a general reader. If its not intelligible to a specialist then there is a problem. There are numerous typos. Again, taking the abstract as an example (e.g. should be 'close' rather than 'closed' in '...closed to the iron-magnetite redox buffer...'). And tense (past / present etc) should be consistent throughout the MS: '...secondary alteration phases formed at relatively low temperatures...', '...CV chondrites witness different physicochemical...'.
→ *OK our manuscript has been checked by a native English speaker colleague.*

One again, we acknowledge the reviewers for their suggestions that clarified, precise and more generally improved considerably this manuscript.

C. Ganino and G. Libourel

Reviewers' Comments:

Reviewer #1:

Remarks to the Author:

I think that this manuscript is significantly improved. The authors have done a thorough job addressing all of the concerns raised in my earlier review, both in their response and where appropriate in the manuscript. I appreciate the detailed response and text additions concerning temperature estimates based on organics and chondrule mineralogy, versus those modeled in the current work (Lines 320-395). The x-ray element maps provided at high-resolution are much more informative and more able to be related to the detailed petrographic descriptions in text. Edits and grammatical corrections made by the authors have resulted in a manuscript that reads clearly. As such, I believe that this manuscript is now suitable for publication and will be of interest to the community.

Respectfully,
kieren.

Reviewer #3:

Remarks to the Author:

I note that all the reviewers asked a similar question regarding the relative level of metamorphism experienced by Allende, and the reduced CVs. Put simply, the reviewers suggestion is that the level of metamorphism experienced by some of these rocks (e.g. Reviewer 2 L204) is not consistent with the authors model, and the relative intensity of metamorphism across the suite is the reverse of that indicated by a range of data from other studies. The authors response to the question is indirect, essentially that different minerals or mineral assemblages may record a range of temperatures at different points during prograde or retrograde metamorphism within the CV parent body. Which seems suggest that its impossible to identify peak metamorphic T in CVs. But the authors do imply it - in stating that the phases observed 'in Vigarano, Leoville or Efremovka, are amongst those requiring the highest temperatures'. That statement says to me that these meteorites experienced something close to peak T in the CV parent body. The obvious implication is that they saw higher temperatures than Allende. It is not simply down to a smooth spectrum recording a range of prograde and retrograde conditions. In addition, while terrestrial metamorphic rocks do indeed record disequilibrium conditions, the entire field is built on the ability to identify peak metamorphic temperature. Why is that any different for CVs?

I like this paper, but this is a significant issue, and the authors need to do a better job of addressing it. It folds into the grain size question - the simple explanation being that Allende saw high T for longer than the other meteorites, so its matrix is partially recrystallised. And it folds into the point about primordial components (which I did not define clearly enough). All data points to reduced CVs experiencing significantly lower temperatures than Allende. This is demonstrated clearly in the abundance of presolar grains. Bulk noble gases, noble gases in diamond, presolar diamond and SiC all point to indicate temperature increasing from Leoville to Vigarano to Allende. These meteorites did not experience the same range of temperatures, the explanation being that 'different phases or phases assemblages may record different temperatures of formation, as on Earth in the case of a prograde or retrograde metamorphism.' And Vigarano and Leoville did not see higher temperatures than Allende. All evidence indicates that these rocks saw significantly lower peak T.

General response

One of the main conclusions of this manuscript is to show that, from the study of the Ca-Fe-rich secondary phases of CV3 chondrites, the classical distinction between reduced and oxidized CV chondrites is no longer valid and that CV3 chondrites experienced instead similar reduced metamorphic conditions closed to the iron-magnetite (IM) redox buffer at low silica activity, i.e., $\log a_{\text{SiO}_2} < -1$ when Ca-Fe –rich minerals formed at moderate temperature (from 210°C and up to 610°C).

It is remarkable that a recent and independent study on the insoluble organic matter's (IOM's) sulfur speciation and structural order (Bose et al., Meteorit. Planet. Sci., 2017, now added to references) propose the formation of secondary phases in Allende (which is classified as CV_{OxA}) at temperature as high as 624°C and in reduced redox conditions. Both T and fO₂ being very similar to our estimates made from inorganic materials give further credit to our finding and the general scenario we propose.

Point to point reply to reviewer :

Reviewer #1 (Remarks to the Author):

I think that this manuscript is significantly improved. The authors have done a thorough job addressing all of the concerns raised in my earlier review, both in their response and where appropriate in the manuscript. I appreciate the detailed response and text additions concerning temperature estimates based on organics and chondrule mineralogy, versus those modeled in the current work (Lines 320-395). The x-ray element maps provided at high-resolution are much more informative and more able to be related to the detailed petrographic descriptions in text. Edits and grammatical corrections made by the authors have resulted in a manuscript that reads clearly. As such, I believe that this manuscript is now suitable for publication and will be of interest to the community.

Respectfully,
kieren.

Thank you for this review. Your comments during the first review clearly improved the manuscript and clarified our conclusion.

Reviewer #3 (Remarks to the Author):

I note that all the reviewers asked a similar question regarding the relative level of metamorphism experienced by Allende, and the reduced CVs. Put simply, the reviewers suggestion is that the level of metamorphism experienced by some of these rocks (e.g. Reviewer 2 L204) is not consistent with the authors model, and the relative intensity of metamorphism across the suite is the reverse of that indicated by a range of data from other studies.

This remark point to the formation of hydrous phases and we might have been not clear in our reply to the first review. Shortly : abundant phyllosilicates observed in Kaba are (1) not very abundant (1.6%, Howard et al, 2010), and (2) not incompatible with our T-aSiO₂ path. One of the main finding of this work is to show that the stability of secondary phases in CV chondrites are controlled by a complex set (multivariant space) of intensive variables, including P, T and the chemical potentials of the system. In this frame, we have shown that the silica activity, aSiO₂ and the fugacity of oxygen, fO₂ are playing a key role. Due to the interplay of these parameters, this means that a phase or a phase assemblage could be stable over a range of these parameters. Fig. 3 shows that hydrous secondary phases can be stable both at low temperature for a low or very silica activity of the system (log aSiO₂ << -1), and in contrast can occur at relatively high temperature for a higher silica activity.

While beyond the scope of this paper devoted to secondary Ca-Fe rich phases in CV matrices, the interplay of these parameters explains well why anthophyllite and talc (could have) formed in chondrule mesostasis at a higher temperature (Brearley, 1997) than in matrices due to the ubiquitous occurrence of olivine and low-Ca pyroxene that buffer during the “alteration” of the chondrule mesostasis the local silica activity according to Mg₂SiO₄ (Fo) + SiO₂(Qtz) = Mg₂Si₂O₆(En) chondrule (see Fig. 9 below in Klein et al., GCA, 2009).

Fig. 9. Temperature-SiO₂ activity plot depicting the phase relations in the system MgO-SiO₂-H₂O. Note the pressure is 200 MPa

In matrices on the other hand, which is overwhelmingly dominated by (ferroan) olivines, the silica activity is not buffered and is simply controlled by the

composition of the fluid phase and/or its level of interaction with the silica activity un-buffered matrix. In this case and as shown in our Fig. 3 for the FeO-SiO₂-H₂O system or in the above figure for the MgO-SiO₂-H₂O system, hydrous phases (e.g., saponite or serpentine-like) could crystallize but only at a well lower temperature. From this example it is thus clear that different phases or phases assemblages may record different temperatures of formation, as on Earth in the case of a prograde or retrograde metamorphism. The same can be said about the evolution of the organic matter in the T-fO₂ space.

The authors response to the question is indirect, essentially that different minerals or mineral assemblages may record a range of temperatures at different points during prograde or retrograde metamorphism within the CV parent body. Which seems suggest that its impossible to identify peak metamorphic T in CVs. But the authors do imply it - in stating that the phases observed 'in Vigarano, Leoville or Efremovka, are amongst those requiring the highest temperatures'. That statement says to me that these meteorites experienced something close to peak T in the CV parent body. The obvious implication is that they saw higher temperatures than Allende. It is not simply down to a smooth spectrum recording a range of prograde and retrograde conditions. In addition, while terrestrial metamorphic rocks do indeed record disequilibrium conditions, the entire field is built on the ability to identify peak metamorphic temperature. Why is that any different for CVs?

We confirm that: as far, it remains challenging to identify peak temperatures for CV. On the one hand, the pertinent phases or assemblages to decipher peak T may have not been identified. On the other hand, the partial or fully retrogression of peak assemblages to lower grade assemblages is very likely here because of the supposed abundance of fluid that promote metamorphic reaction all along the T-aSiO₂ path. These two issues are illustrated in terrestrial metamorphic petrology, the peak conditions being re-evaluated to higher grades when new high grade phases are discovered and the retrogression unfortunately erasing part or the totality of the peak conditions, especially when fluid are present (e.g. White and Powell, 2008, J. Met. Pet.).

I like this paper, but this is a significant issue, and the authors need to do a better job of addressing it. It folds into the grain size question - the simple explanation being that Allende saw high T for longer than the other meteorites, so its matrix is partially recrystallised. And it folds into the point about primordial components (which I did not define clearly enough). All data points to reduced CVs experiencing significantly lower temperatures than Allende. This is demonstrated clearly in the abundance of presolar grains. Bulk noble gases, noble gases in diamond, presolar diamond and SiC all point to indicate temperature increasing from Leoville to Vigarano to Allende. These meteorites did not experience the same range of temperatures, the explanation being that 'different phases or phases assemblages may record different temperatures of formation, as on Earth in the case of a prograde or retrograde metamorphism.' And Vigarano and Leoville did not see higher temperatures

than Allende. All evidence indicates that these rocks saw significantly lower peak T.

This reviewer is right. Even if we have stated that high temperatures could be possibly reached in well-localized area of CV chondrite matrices, we don't have any possibility from our survey of Ca-Fe-rich parageneses to sort each of CV3 chondrites according to their respective peak metamorphic temperature. Therefore, we added in the manuscript a reference to presolar grains and modified the text (in the section "Are these conditions realistic with respect to other observables from CV3 chondrites? I) temperature") accordingly to show that our model is also consistent with the classical order of metamorphic temperature inferred from presolar grain abundances and Raman spectroscopy studies of organic material, i.e., temperature increasing from Leoville to Vigarano to Allende.

We would like however to draw the attention of this reviewer on the fact that presolar grain abundance study, noble gas analysis, as well as spectroscopy study of organic matter are all based on bulk-rock analyses that provide averaged information, e.g., T, t, redox, etc, for CV3 chondrites. That Ca-Fe-rich phases and carbonaceous matter (Bonal et al., 2016) indicate a range of temperature extending over several hundreds of degrees suggests that thermal heterogeneity is a characteristic feature of matrix of CV chondrites. As commonly studied on Earth in metamorphic terrains, it was therefore an objective of this work (see our introduction) to show that the petrological and thermodynamical survey of the various parageneses present in well localized area of CV chondrite matrices could provide a more detailed description of the conditions (including possible lower min. and higher max. T values) reigning in their parent body.

In the light of our inference of reducing conditions in the CV3 parent body, and as already mentioned, which is also supported in the case of Allende (CV_{OxA}) by such recent study on the insoluble organic matter's (IOM's) sulfur speciation and structural order (Bose et al. 2017), it could be thus interesting to reconsider i) the fate of presolar grains, like SiC, graphite or diamond in CV chondrites, which are obviously more stable in reducing conditions than in oxidizing ones as well as ii) the significance of their abundances in term of thermal history.